# Reference-free assembly of long-read transcriptome sequencing data with RNA-Bloom2

Ka Ming Nip [1,2] ✉, Saber Hafezqorani [1,2], Kristina K. Gagalova [1,2], Readman Chiu [1], Chen Yang [1,2], René L. Warren [1] & Inanc Birol [1,3] ✉

Long-read sequencing technologies have improved significantly since their emergence. Their read lengths, potentially spanning entire transcripts, is advantageous for reconstructing transcriptomes. Existing long-read transcriptome assembly methods are primarily reference-based and to date, there is little focus on reference-free transcriptome assembly. We introduce "RNA-Bloom2 [https://github.com/bcgsc/RNA-Bloom]", a reference-free assembly method for long-read transcriptome sequencing data. Using simulated datasets and spike-in control data, we show that the transcriptome assembly quality of RNA-Bloom2 is competitive to those of reference-based methods. Furthermore, we find that RNA-Bloom2 requires 27.0 to 80.6% of the peak memory and 3.6 to 10.8% of the total wall-clock runtime of a competing reference-free method. Finally, we showcase RNA-Bloom2 in assembling a transcriptome sample of *Picea sitchensis* (Sitka spruce). Since our method does not rely on a reference, it further sets the groundwork for large-scale comparative transcriptomics where high-quality draft genome assemblies are not readily available.

RNA sequencing (RNA-seq) has become the standard method for gene/transcript discovery, transcriptome profiling, and isoform expression quantification. Since the dawn of high-throughput short-read sequencing technologies, transcriptome assemblies have enabled the discovery of novel isoforms[1], identification of foreign RNAs[2], intra-species gene-fusion transcripts[3] and inter-species chimeric transcripts[4,5], and guided scaffolding[6] and annotation of draft genome assemblies. Such applications have been key in enhancing our understanding of genome biology and the etiology and progression of various diseases.

Pacific Biosciences of California, Inc. (PacBio, Menlo Park, CA) and Oxford Nanopore Technologies PLC (ONT, Oxford, UK) have been offering long-read sequencing technologies commercially since 2011 and 2014, respectively. Both sequencing technologies have improved significantly since their emergence to yield increased read length, base

accuracy, and throughput[7]. In particular, ONT's MinION devices are small and portable, thus having the potential to allow rapid sequencing and downstream analyses[8]. Moreover, nanopore sequencing enables direct RNA (dRNA) sequencing without the need to generate complementary DNA (cDNA) libraries[9]. On the other hand, PacBio's single-molecule-real-time (SMRT) sequencing provides circular consensus sequencing (CCS) to produce reads that have a lower base error rate than that of ONT reads[10]. As a result, the number of computational methods designed for processing and analyzing long-read sequencing data is growing rapidly[7].

Compared to Illumina short-read sequencing technologies, long reads are noisier but are several orders of magnitude longer, making them able to span through multiple exons and even capture full-length transcripts in some instances, thus simplifying the transcriptome assembly problem. However, existing transcriptome assemblers, such

[1]Canada's Michael Smith Genome Sciences Centre, BC Cancer, Vancouver, BC V5Z 4S6, Canada. [2]Bioinformatics Graduate Program, University of British Columbia, Vancouver, BC V5Z 4S6, Canada. [3]Department of Medical Genetics, University of British Columbia, Vancouver, BC V6T 1Z3, Canada. ✉e-mail: kmnip@bcgsc.ca; ibirol@bcgsc.ca

as StringTie2[11], are predominantly reference-based where transcripts are derived from spliced-alignment of reads against the reference genome. Genome annotations contain rich information about gene structures that may be utilized for guiding reference-based transcriptome assembly; some examples include: refinement of spliced-alignment based on known splice junctions, inference of transcript strand based on annotated gene orientation, and resolution for antisense transcripts based on known transcription start and end sites. Consequently, a subclass of reference-based assemblers, such as FLAIR[12], require a genome annotation in addition to the reference genome for accurate isoform reconstruction. Similarly, StringTie2 has the option of incorporating transcriptome annotation to aid its assembly process.

Reference-free assembly of transcriptomes is especially valuable when there is no available reference genome or the reference genome is still at the draft stage, which may not fully support the reference-based assembly of all transcripts in a given transcriptome sequencing sample. In general, long-read reference-free genome assembly algorithms such as wtdbg2[13] (also known as Redbean) are not suitable for transcriptome data because they cannot reconstruct alternative isoforms and they typically assume a uniform sequencing depth, which is practically nonexistent in transcriptomic data due to varying transcript expression levels. Reference-free assembly methods typically rely on read-to-read mapping, whereas reference-based methods rely on read-to-reference alignments. Since read-to-read mapping is much more resource-intensive than read-to-reference alignment, reference-free methods tend to have a much higher computational cost than reference-based methods. To find wider applications, reference-free assembly algorithms need to overcome the challenges in managing computational resources.

Sequence clustering-based assembly follows the divide-and-conquer paradigm and thus it requires less resources than methods that align all reads against each other. RATTLE[14] is an example of such a method, and to the best of our knowledge, it is the only reference-free transcriptome assembler that can assemble transcripts solely from long-read sequencing data. RATTLE clusters input reads into isoform-based (or gene-based) groupings and derives consensus sequences from each read cluster to reconstruct full-length transcripts. Nevertheless, clustering accuracy is an important factor in assembly quality and computational performance. A lenient clustering criterion would create few but large read clusters, resulting in slow runtime, high peak memory, and aggregation of reads from too many genes. A stringent clustering criterion, on the other hand, would create many small clusters, potentially resulting in insufficient aggregation of reads and incomplete transcript reconstruction.

Digital normalization[15], also known as in silico read normalization, is a simple but effective method to improve the computational performance of reference-free assemblers by reducing the number of overrepresented reads, such as those of high-expressed transcripts, based on the saturation of $k$-mers in the reads. In contrast to naive subsampling, digital normalization is better at preserving low-expression transcripts. However, it has been primarily utilized for the assembly of short-read RNA-seq data[16,17]. With the introduction of strobemers[18] as a mismatch and indel tolerant alternative to $k$-mers, digital normalization with strobemers should be highly applicable to transcriptome assembly of noisy long reads.

Here we present RNA-Bloom2, the successor to our short-read transcriptome assembly tool, RNA-Bloom[19], that extends support for reference-free transcriptome assembly of bulk RNA long sequencing reads. RNA-Bloom2 offers both memory- and time-efficient assembly by utilizing digital normalization of long reads with strobemers. Our benchmarking shows that RNA-Bloom2 requires 27.0–80.6% of the peak memory and 3.6–10.8% of the total wall-clock runtime of RATTLE. In simulated datasets, RNA-Bloom2 has 1.5–7.3% higher recall and 0.3–1.5% lower false discovery rates than RATTLE. In the spike-in

datasets, RNA-Bloom2 has 5.7–17.6% higher recall than RATTLE. Finally, we showcase RNA-Bloom2 in assembling a transcriptome sample of *Picea sitchensis* (Sitka spruce), without using a genomic reference.

## Results

### Reference-free transcriptome assembly with RNA-Bloom2
RNA-Bloom2's six-stage workflow for the reference-free transcriptome assembly of long reads is summarized in Fig. 1. In stage one, long reads are corrected for errors in an alignment-free approach based on a Bloom filter de Bruijn graph of $k$-mers derived from input reads. Short reads can be optionally provided to aid in the error correction of long reads. In stage two, the set of corrected reads is digitally normalized with strobemers, such that overrepresented reads are removed to yield a target read depth. Stages one and two are highly integrated to reduce input-output operations. Since only a portion of corrected reads would be retained by digital normalization, stage one is not meant to exhaustively correct all errors in the reads and is instead intended to be fast and memory-efficient. In stage three, reads in the normalized set are overlapped against each other to identify low-depth regions in the reads to be trimmed or split. In stage four, trimmed reads are overlapped against each other to generate an overlap graph where reads on each unambiguous path are assembled into a "unitig". In stage five, the unitigs, which may still contain errors, are polished using the alignments of corrected reads from stage one. In stage six, the polished unitigs are aligned against each other to generate an overlap graph where transcripts are derived based on the length-normalized read depth of the unitigs. If the reads are produced by the cDNA sequencing protocol, sequences containing potential poly(A) tails are identified in order to prune the overlap graphs in stages four and six. A more detailed description of each stage is provided in the Methods section. It is important to note that all-versus-all sequence comparisons in RNA-Bloom2 are performed after digital normalization, as opposed to after sequence clustering as in RATTLE. Since the number of sequences retained after digital normalization is expected to be much lower than the number of raw input reads, sequence clustering is not necessary for RNA-Bloom2.

### Evaluation of error correction and digital normalization
We evaluated the effectiveness of the error correction and digital normalization stages of RNA-Bloom2 using experimental data. We selected one mouse dataset from the Long-read RNA-Seq Genome Annotation Assessment Project (LRGASP) Consortium[20] containing the matching sequencing data for ONT cDNA, ONT dRNA, PacBio CCS, and Illumina reads of the same biological sample (Supplementary Table 1). ONT dRNA and PacBio CCS reads do not contain adapters, but ONT cDNA reads and Illumina reads are trimmed for adapters with Pychopper[21] and Trimmomatic[22], respectively (Supplementary Method 1). Out of the three long-read samples, the ONT cDNA sample has the largest number of sequencing reads (13,127,667 reads) but the lowest read alignment rate (78.66%) against the combined reference genome for mouse GRCm39 and Lexogen's Spike-In RNA Variant (SIRV) transcripts[23] from the LRGASP consortium (Synapse accession "syn25683365"). Compared to the ONT cDNA sample, the ONT dRNA and PacBio CCS samples have only one-sixth of the reads (2,153,439 reads and 2,144,172 reads, respectively) but higher read alignment rates (95.58% and 95.49%, respectively) against the reference genome. The Illumina sample has 40,225,298 read pairs (2 × 100 nucleotides (nt)) and is only used for the hybrid error correction of the long reads in RNA-Bloom2.

We first assess both methods of alignment-free error correction in RNA-Bloom2: (i) using only long reads, and (ii) using a hybrid of long and short reads. We investigated the nucleotide base error rates of the reads before and after error correction (Supplementary Table 2); error

rates are measured by Trans-NanoSim[24] (Supplementary Method 2). The error rates of the reads before error correction in the ONT dRNA, ONT cDNA, and PacBio CCS samples are 12.17%, 7.18%, and 1.96%, respectively. Long-read-only error correction has reduced the error rates to 10.28%, 4.03%, and 1.35% for the ONT dRNA, ONT cDNA, and PacBio CCS samples, respectively. As expected, hybrid error correction has resulted in even lower error rates of 6.55%, 3.51%, and 1.34% for the ONT dRNA, ONT cDNA, and PacBio CCS samples, respectively. Long-read-only error correction has the largest reduction (−3.15%) in the error rate in the ONT cDNA sample, whereas hybrid error correction has the largest reduction (−5.62%) in the error rate in the ONT dRNA sample.

We next investigated the percentage of reads remaining after digital normalization and the percentage of input reads aligned to the final assembly (Supplementary Table 3, Supplementary Method 3). For assemblies with long-read-only error correction, 48.15%, 3.76%, and 11.66% of reads remained after digital normalization in the ONT dRNA, ONT cDNA, and PacBio CCS samples, respectively. For assemblies with hybrid error correction, 38.80%, 3.53%, and 11.63% of reads remained after digital normalization in the ONT dRNA, ONT cDNA, and PacBio CCS samples, respectively. The ONT dRNA assemblies have the highest percentages of reads remaining after digital normalization. This is likely due to the much higher error rate in the reads, which limits the number of matching strobemers among the reads. Despite the fact that a substantial proportion of reads are removed by digital normalization, 97.44–97.54% and 95.04–95.16% of input reads are still able to align to the final assemblies for the ONT dRNA and PacBio CCS samples, respectively. Although 73.52–74.02% of input reads aligned the final assembly for the ONT cDNA sample, it is important to note that only 78.66% of reads in the sample were aligned against the reference genome. These results confirm that digital normalization in

RNA-Bloom2 is effective in removing overrepresented reads from long-read transcriptome sequencing data.

Finally, we assess how well digital normalization preserves the number of genes represented in reads, using the same experimental samples as before. For a given set of reads, the number of represented genes is defined as the number of genes with expression levels above zero (Supplementary Method 9). The results are summarized in Supplementary Table 4. Overall, the number of genes represented in the digitally normalized reads is 95.6–98.3% of those in the raw reads. It is important to note that error correction already lowers the gene representation (due to filtering of reads with low *k*-mer multiplicities) prior to digital normalization. In general, the majority of genes discarded are very lowly expressed, and their expression levels are predominantly lower than those of genes retained (See Supplementary Table 13). Since only a small proportion of reads remains after digital normalization, the 1.7–4.4% reduction in gene representation is within our expectations.

## Assembly benchmarking with simulated datasets

We benchmarked the assembly quality and the computational performance of RNA-Bloom2 on simulated data. Assembly quality is measured based on the metrics described in Table 1. The assembly evaluation procedure is described in the Methods section. We prepared two mouse-simulated datasets with Trans-NanoSim[24] for the cDNA and dRNA sequencing protocols modeled on experimental ONT data (See Methods). To investigate the effect of sequencing depth, we subsampled each dataset to 2, 10, and 18 million reads, resulting in a total of six sets of reads for our benchmarking experiments. The features of the simulated datasets are presented in Supplementary Table 5. Compared to the cDNA dataset, the dRNA dataset has a higher error rate, longer N50 read length, and fewer simulated transcripts.

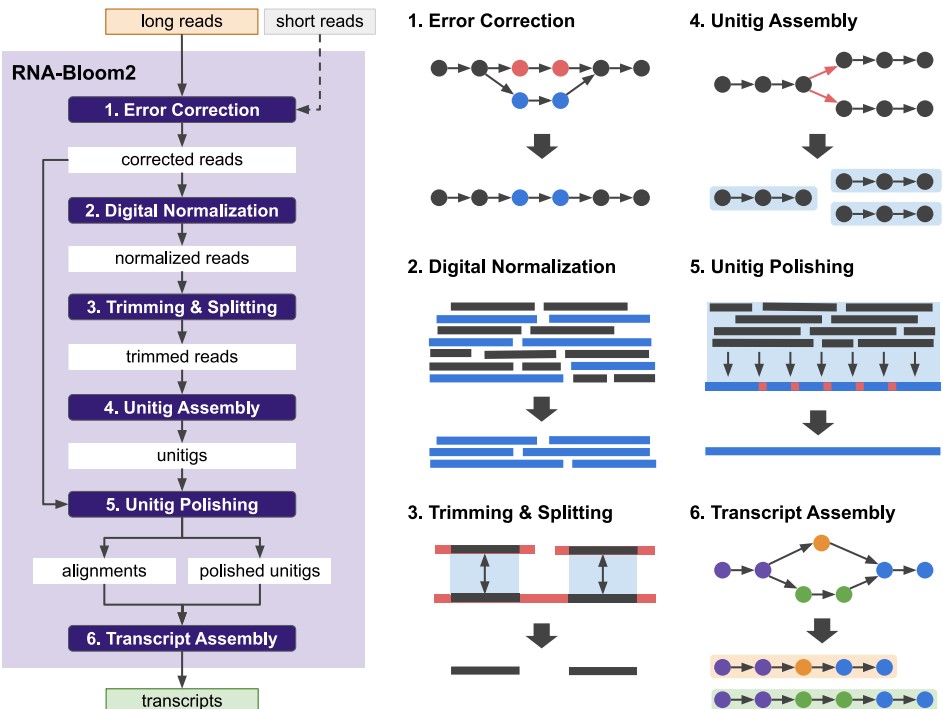

**Fig. 1 | RNA-Bloom2 assembly workflow overview.** The long-read assembly workflow of RNA-Bloom2 consists of six stages. In stage 1, input long reads are corrected for errors using a de Bruijn graph, where erroneous *k*-mers (red circles) are replaced with *k*-mers of higher multiplicities (blue circles). The de Bruijn graph can optionally include *k*-mers from short reads if provided. In stage 2, corrected reads are digitally normalized to select a subset of reads (blue rectangles). In stage 3, normalized reads are trimmed and splitted at regions of low read depth (red rectangles). In stage 4, an overlap graph of trimmed reads is generated to assemble unambiguous paths into unitigs (blue paths). In stage 5, residual errors (red squares) in unitigs are polished using alignments of corrected reads from stage 1. In stage 6, an overlap graph of polished unitigs is generated to assemble alternative transcripts (orange and green paths).

**Table 1 | Transcriptome assembly quality assessment metrics**

| Metric | Definition |
|---|---|
| Complete reconstruction | Truth set transcript reconstructed at least 95% in length |
| Partial reconstruction | Truth set transcript reconstructed between 0 and 95% in length |
| Missing reconstruction or false negative (FN) | Truth set transcript with no detectable reconstruction |
| True positive (TP) | Truth set transcript with complete or partial reconstruction |
| False positive (FP) | Reference transcript not in the truth set |
| Misassembly (MA) | Incorrectly assembled contig with segments from one or more reference transcripts. It can be intragenic or intergenic. |
| Intragenic misassembly | Incorrectly assembled contig with segments from reference transcripts of the same gene |
| Intergenic misassembly | Incorrectly assembled contig with segments from reference transcripts of different genes |
| Large indel (LI) | Contigs with at least one large indel (> 70 nt) compared to the best-aligning reference transcript. Large indels can arise from sequencing errors, alternative donors/acceptors, skipped exons, and retained introns. |
| Unclassified (UC) | Contigs with no alignments against reference transcripts. These contigs can arise from noise or incorrect error correction. |
| Recall | Percentage of truth set transcripts reconstructed. |
| False discoveries (FD) | = FP + MA + LI + UC |
| False discovery rate (FDR) | = FD / (FD + TP) |
| F1 | = TP / [TP + 0.5 * (FD + FN)] |
| Redundancy | = (contigs representing TP) / TP |

These metrics are intended for sequencing data with a known ground truth where true-positives and false-positives can be easily discerned. The truth set transcripts are either the set of simulated transcripts or the set of spike-in transcripts in real data.

Using the two-million simulated read sets, we illustrate the value of reference-free transcriptome assembly by comparing RNA-Bloom2 against isONcorrect[25], which is an error-correction tool designed for ONT transcriptome data. Commands for running both tools are described in Supplementary Method 10. We evaluated the number of complete transcripts and false discoveries detected for both tools (Supplementary Fig. 1). For cDNA data, output reads of isONcorrect have 7104 complete transcripts and 4852 false discoveries, whereas the RNA-Bloom2 assembly has 8929 complete transcripts and 4028 false discoveries. For dDNA data, output reads of isONcorrect have 7646 complete transcripts and 2606 false discoveries, whereas the RNA-Bloom2 assembly has 11,157 complete transcripts and 2559 false discoveries. Overall, RNA-Bloom2 offers 25.7–45.9% increase in complete transcripts and 1.8–17.0% reduction in false discoveries compared to isONcorrect. Although long reads often span multiple exons, our results show that transcriptome assembly is valuable to reference-free analysis of long-read transcriptome data.

We then compared RNA-Bloom2 against three other transcriptome assembly tools designed for long reads: RATTLE, StringTie2, and FLAIR. RATTLE is the only other reference-free method, whereas StringTie2 and FLAIR are entirely reference-based. In addition, all FLAIR assemblies were guided by the reference transcriptome annotation in conjunction with the associated reference genome. StringTie2 was run in two different modes: with and without the transcriptome annotation, which are denoted as "StringTie2_GTF" and "StringTie2" from hereon. Since reference-based methods are expected to perform better than reference-free methods, StringTie2, StringTie2_GTF, and FLAIR serve as the baseline for evaluating the performance of RNA-Bloom2 and RATTLE. All assembly methods were run with 48 threads using the same compute nodes with the exception of FLAIR and RATTLE for the assemblies of the 18 million-read sets, which were reprocessed on a high-memory machine after failing the initial runs. The assembly evaluation procedure is described in the Methods section. Commands for all methods and computing hardware are documented in Supplementary Method 6.

The computational performance of all five assembly methods is summarized in Fig. 2 and Supplementary Tables 6 and 7. StringTie2 and StringTie2_GTF performed similarly and they have the fastest runtimes and consistently low peak-memory usage for all datasets.

FLAIR has the worst peak-memory usages and RATTLE has the worst total runtimes. As expected, reference-based assemblers are faster than reference-free assemblers. RNA-Bloom2 has the lowest memory usage for the 2 million-read cDNA dataset. The peak memory usage and total runtimes of RATTLE are 1.24–3.70 and 9.22–28.12 times of those of RNA-Bloom2, respectively. Both RNA-Bloom2 and RATTLE require a higher peak-memory usage in assembling the dRNA datasets than the cDNA datasets, possibly due to the higher error rate and higher N50 read length of the dRNA datasets. However, the peak memory of RNA-Bloom2 for the dRNA datasets did not increase exponentially with respect to the number of input reads. This suggests that the digital normalization stage in RNA-Bloom2 is effective in reducing the number of reads because the number of transcripts in the 10 million-read set and the 18 million-read set only differs by 210 (Supplementary Table 5).

The benchmarking results for simulated data are presented in Fig. 3. The trends for recall are similar for both simulated cDNA and dRNA datasets (Fig. 3a). RNA-Bloom2 has higher percentages (+1.5 to +7.3%) of complete reconstruction than RATTLE in all simulated samples. The largest difference is observed in the 18 million-read cDNA sample, whereas the smallest difference is observed in the 2 million-read cDNA sample. RNA-Bloom2 also has lower percentages (−9.2 to −11.9%) of missing transcripts than RATTLE in all samples. Behind StringTie2_GTF and FLAIR, RNA-Bloom2 has the third smallest percentages of missing transcripts (41.3–57.0% for cDNA sets and 24.5–49.5% for dRNA sets). For both cDNA and dRNA datasets, StringTie2_GTF has the highest percentages of complete reconstruction (46.2–79.9%) and the smallest percentages of partial reconstruction (4.3–11.4%). However, StringTie2 has the highest percentage of missing transcripts in all cDNA (57.4–68.5%) and dRNA (41.8–56.8%) samples.

We further investigated assembly recall with respect to transcript expression levels. We assigned simulated transcripts to expression quartiles: low, medium-low, medium-high, and high. The expression-stratified assembly recall results for simulated cDNA and dRNA datasets are presented in Supplementary Figs. 2 and 3, respectively. StringTie2_GTF has the most complete reconstruction in all four expression quartiles for both cDNA and dRNA datasets. RNA-Bloom2 has higher percentages of complete reconstruction than RATTLE in the

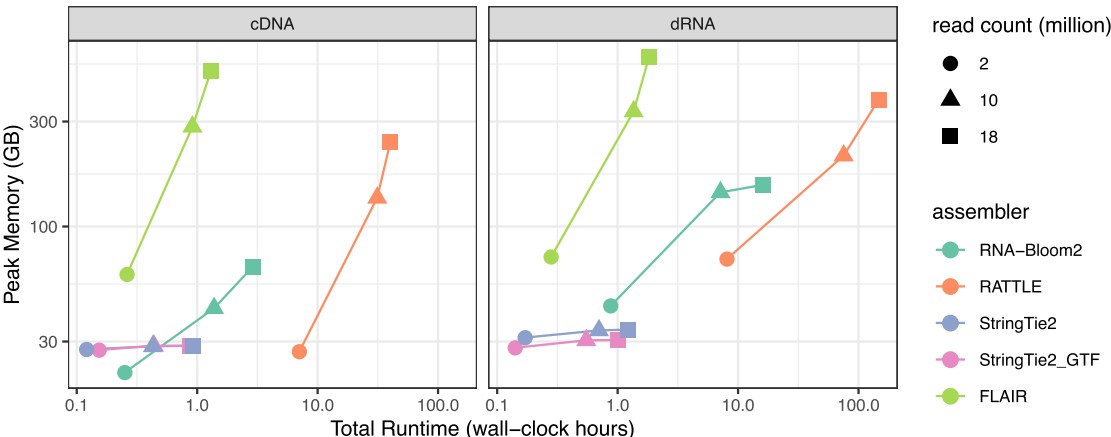

**Fig. 2 | Computational performance on simulated datasets.** All assemblers were run with 48 threads and assemblies were generated for 2, 10, and 18 million simulated reads of cDNA and dRNA samples. Peak memory usage was measured in gigabytes (GB), and runtime was measured in wall-clock hours. Both axes are in logarithmic scale. For StringTie2, StringTie2_GTF, and FLAIR, performance figures include read alignment and generation of indexed BAM files. Source data are provided as a Source Data file.

high and medium-expression expression quartiles, with the exception of the 2 million-read cDNA sample. However, the opposite was observed in the low quartiles. In the high expression quartile, FLAIR has the lowest percentage of complete reconstruction except for the 2 million-read cDNA sample, but it has the second-lowest percentage of missing reconstruction. In the low expression quartiles, RNA-Bloom2 has the lowest percentages of complete reconstruction, but both annotation-guided approaches, StringTie2_GTF and FLAIR, are the two best-performing methods.

We also evaluated false-discovery rates (FDR), F1, and redundancy (Fig. 3b–d) for the five methods. RNA-Bloom2 has lower FDR (−1.3 to −1.5% for cDNA sets, −0.3 to −1.0% for dRNA sets) compared to RATTLE across all samples. StringTie2 has the highest FDR (9.6 to 14.2% for cDNA sets, 8.9 to 12.9% for dRNA sets), but the FDR of StringTie2_GTF are much lower (2.7–3.5% for cDNA sets, 3.1 to 6.5% for dRNA sets). While StringTie_GTF has much reduced mis-assemblies and large indel contigs compared to StringTie2, its false discoveries are primarily contributed by false positive reference transcripts. FLAIR has the lowest FDR in all simulated samples (0.2–0.6%). The F1 scores of RNA-Bloom2 are higher (+9.2 to +10.0% for cDNA sets, +7.5 to +10.5% for dRNA sets) than those of RATTLE. StringTie2_GTF has the highest F1 scores (72.3–83.6% for cDNA sets, 81.3–88.4% for dRNA sets), whereas StringTie2 has the lowest F1 scores (46.6–56.5% for cDNA sets, 58.4–68.8% for dRNA sets). RNA-Bloom2 has lower redundancy (−0.2 to −0.7 for cDNA sets, −0.1 to −0.4 for dRNA sets) than RATTLE.

StringTie2_GTF and StringTie2 have the lowest redundancy (1.0–1.1 and 1.1–1.3, respectively), while FLAIR and RATTLE have the highest redundancy (1.3–1.8 and 1.2–1.9, respectively). The high redundancy of FLAIR and RATTLE is potentially a result of larger numbers of contigs assembled (Supplementary Table 11). However, the number of contigs per gene is similar to the number of transcripts per gene in the ground truth for multi-transcript genes (Supplementary Fig. 9).

Finally, we assess the isoform assembly precision based on the transcript models in the simulated data. The precision metric is calculated using the isoform classification from SQANTI3[26] (see Methods for details). Our results are summarized in Supplementary Fig. 8. StringTie2_GTF and FLAIR have the highest isoform assembly precision (87.1–93.6% and 84.7–92.4%, respectively). RNA-Bloom2 has the lowest precision (75.9%) in the 18 M dRNA sample while RATTLE has the lowest precision (76.6–81.0%) in all other samples, where RNA-Bloom2's precisions are +2.1 to +9.2% higher.

## Assembly benchmarking with spike-in control data

In addition to simulated data, we also benchmarked the four assembly methods on experimental sequencing data of known sequences. We selected one mouse dataset from the LRGASP Consortium containing the matching sequencing data for ONT cDNA, ONT dRNA, and PacBio CCS of the same biological sample. The sequencing samples for this dataset were spiked with Lexogen's Spike-In RNA Variant (SIRV) transcripts[23] containing 92 External RNA Control Consortium (ERCC) spike-ins, 69 SIRV isoforms, and 15 long SIRVs. We extracted the reads corresponding to the spike-ins (See Methods) for assembly bench-marking and the features of the spike-in datasets are summarized in Supplementary Table 8. The PacBio CCS sample has the longest N50 read length (2,460 nt) and the lowest error rate (2.03%). The ONT cDNA sample has the shortest N50 read length (712 nt) but the highest number of reads (n = 404,783). The ONT dRNA sample has the fewest reads (n = 26,814) and the highest error rate (11.01%).

We evaluated the assembly quality of the spike-in samples based on the metrics described in Table 1, and the benchmarking results are presented in Fig. 4. The trends for recall are similar across platforms (Fig. 4a). In all three samples, StringTie2_GTF has the highest percentages of complete reconstruction (55.1–76.1%). RNA-Bloom2 has higher percentages of complete reconstruction (+5.7 to +17.6%) than RATTLE in all three samples; RNA-Bloom2 ranks the second highest in both ONT samples. RNA-Bloom2 has the smallest percentage of missing reconstruction in the ONT cDNA sample (25.6%) while StringTie2_GTF has the smallest percentages of missing reconstruction in the ONT dRNA and PacBio CCS samples (42.0% and 23.9%, respectively).

Overall, false discoveries (Fig. 4b) are predominantly attributed to intragenic misassemblies and large indel contigs. RNA-Bloom2 assemblies contain unclassified contigs that cannot be aligned to the reference spike-in transcripts. Unlike the simulated datasets, there are no false-positive reference transcripts because all reference spike-in transcripts are true positives. StringTie2_GTF has the lowest false-discovery rates (FDR) in all three samples (0–4.3%). StringTie2 has the highest FDR in the ONT dRNA and cDNA samples (25.6% and 20.6%, respectively) while RATTLE has the highest FDR in the PacBio CCS sample (16.3%). RNA-Bloom2 has the second highest FDR in the ONT cDNA sample (20.4%). Both FLAIR and RNA-Bloom2 have over seven times higher FDR in the ONT cDNA sample than in the ONT dRNA and PacBio CCS samples. StringTie2_GTF has the highest F1 scores in all samples (73.4–86.5%). RNA-Bloom2 has the second highest F1 scores in ONT dRNA and PacBio CCS samples (64.2% and 80.3%, respectively), but it has the lowest F1 score in the ONT cDNA sample (51.4%). RATTLE

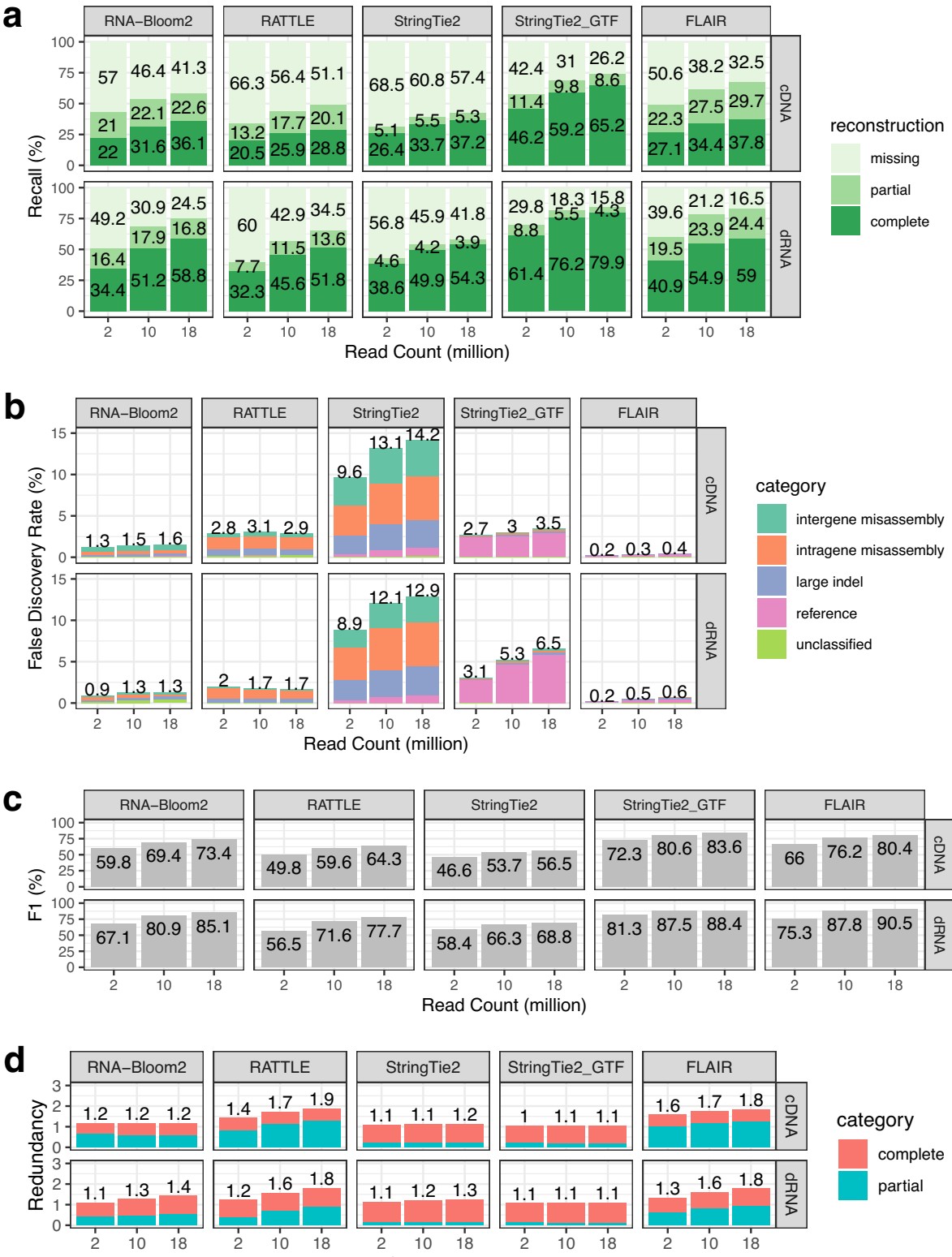

**Fig. 3 | Assembly quality evaluation of simulated datasets. a** Recall is measured with respect to transcription reconstruction levels. **b** False discovery rate is measured based on the numbers of intragene and intergene misassemblies, false positive reference transcripts, contigs with large indels, and unclassified contigs. **c** F1 scores. **d** Redundancy is defined as the number of contigs representing true-positive transcripts with complete or partial reconstruction divided by true-positive transcripts with complete or partial reconstruction. Source data are provided as a Source Data file.

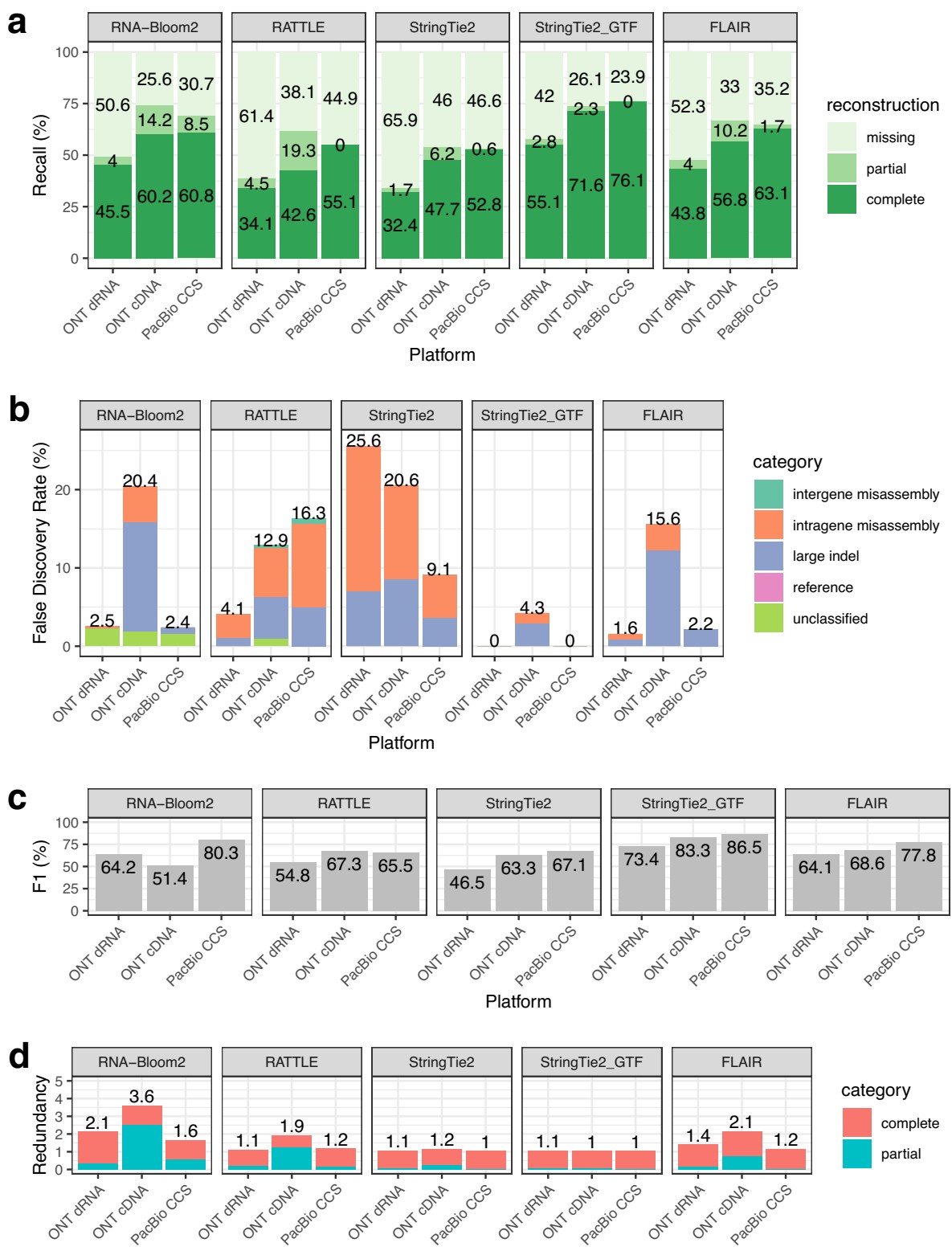

**Fig. 4 | Assembly quality evaluation of spike-in control datasets. a** Recall, **b** false discovery rates, **c** F1 scores, and **d** redundancy were evaluated for each assembly method on spike-in control data generated from three sequencing technologies: ONT direct RNA, ONT cDNA, and PacBio CCS. The spike-in control data were extracted from a mouse dataset from the LRGASP Consortium. Source data are provided as a Source Data file.

has the lowest F1 score in the PacBio CCS samples (65.5%) and StringTie2 has the lowest F1 score in the ONT dRNA sample (46.5%).

StringTie2_GTF and StringTie2 have the lowest redundancy in all samples (1.1–1.2). RNA-Bloom2 has the highest redundancy in all three samples (2.1, 3.6, 1.6 in ONT dRNA, ONT cDNA, and PacBio CCS, respectively), +0.5 to +1.7 higher than RATTLE. In particular, RNA-Bloom2's redundancy is predominantly contributed by partially reconstructed transcripts in the ONT cDNA sample, but its redundancy is predominantly contributed by completely reconstructed transcripts in the ONT dRNA and PacBio CCs samples. The high redundancy of RNA-Bloom2 assemblies is potentially due to the substantially higher number of contigs assembled (Supplementary Table 12). Although the degree of redundancy varies from one method to another, all assembly methods do not follow the number of transcripts per gene in the ground truth (Supplementary Fig. 10). This is different from what was observed in the simulated data, and we speculate that this is a result of the small number of the spike-ins.

### Assembly evaluation in mouse experimental sample replicates

Using each assembly method, we have assembled individual replicates of mouse experimental samples from the LRGASP consortium for ONT cDNA, ONT dRNA, and PacBio CCS platforms. These are the same samples where the spike-in control dataset was extracted (Supplementary Method 5). For PacBio replicates, RATTLE performed gene-based clustering instead of isoform-based clustering, which did not run to completion in our initial attempts. We assess the assemblies with SQANTI3 (Supplementary Method 11) and our results are summarized in Supplementary Figs. 11–13. RNA-Bloom2 recovered a higher number of annotated genes compared to RATTLE in all three platforms. Compared to reference-free methods, reference-based methods have substantially higher numbers of novel genes in ONT cDNA and PacBio CCS platforms. FLAIR has the highest number of known canonical splice junctions in all three platforms; RNA-Bloom2 is a close second in the PacBio CCS platform. Compared to reference-based methods, reference-free methods have substantially higher numbers of non-canonical splice junctions; RNA-Bloom2 has the highest in the ONT cDNA platform. The proportion of non-canonical splice junctions in RNA-Bloom2 is particularly higher than other categories in the ONT cDNA platform (Supplementary Fig. 14). RNA-Bloom2 has the most full-splice match isoforms and incomplete-splice match isoforms. FLAIR has the most novel in-catalog isoforms. RNA-Bloom2 has the most novel not-in-catalog isoforms and genic-genomic isoforms. All methods have few antisense isoforms. StringTie2 has the most antisense isoforms in the ONT cDNA platform, while RNA-Bloom2 has the most antisense isoforms in the other two platforms. Compared to reference-free methods, reference-based methods have substantially higher numbers of intergenic isoforms in ONT cDNA and PacBio CCS platforms. All methods have relatively low numbers genic-intron isoforms; StringTie2 and StringTie2_GTF have the most in all three platforms.

### Reference-free assembly of a Sitka spruce transcriptome

The Sitka spruce (*Picea sitchensis*) is a large, evergreen, and long-living conifer species native to the Pacific Northwest in North America. Although a 20 Gbp draft genome assembly is publicly available[27], its scaffold N50 length is 56.8 kbp, which reflects the draft stage of this short-read genome assembly. In particular, the Sitka spruce's alternative splicing pattern has not been fully investigated. Since conifers are known for their long introns[28,29], the fragmented draft genome would highly limit the effectiveness of reference-based transcriptome assembly methods. Thus, it is a valuable use case to illustrate the utility of reference-free transcriptome assembly methods. Using RNA-Bloom2 and RATTLE, we assembled RNA-seq data from mixed tissue (young needle, bark, xylem, and mature needle) cDNA sampled from a Sitka spruce Q903 spruce weevil-susceptible individual, originated from Haida Gwaii, British Columbia, Canada (53.917, −132.083). The

cDNA sample was sequenced on a ONT MinION device (R9.4 flow cell) and the reads are basecalled with Guppy (See Methods and Supplementary Method 7). A total of 1,323,043 ONT reads with N50 read length of 1,543 nt remained after adapter-trimming with Porechop[30]. We also performed an additional RNA-Bloom2 assembly with hybrid error correction using Illumina paired-end RNA-seq data from a previous study[31].

First, we measured the completeness of single-copy orthologs with BUSCO[32] for the adapter-trimmed ONT reads, the two RNA-Bloom2 assemblies, and the RATTLE assembly (Supplementary Method 7). BUSCO provides a quantitative assessment of expected gene content for each set of transcript sequences, and the results are summarized in Supplementary Table 9. The RNA-Bloom2 assembly with hybrid error correction has the highest percentage of complete BUSCO and the lowest percentages of fragmented and missing BUSCO. Specifically, the complete BUSCO has improved from 73.4% in the adapter-trimmed reads to 87.6% in the RNA-Bloom2 assembly, whereas the percentages of fragmented and missing BUSCO in the reads (7.7% and 18.9%) have reduced by half after assembly with RNA-Bloom2 (3.4% and 9.0%). On the other hand, the RNA-Bloom2 assembly with long-read-only error correction has a higher percentage of complete BUSCO and lower percentages of fragmented and missing BUSCO than the input reads and the RATTLE assembly. Compared to the reads, the RATTLE assembly has a lower percentage of complete BUSCO and higher percentages fragmented and missing BUSCO.

We have selected the RNA-Bloom2 assembly with hybrid error correction for further analyses. This transcriptome assembly has a total of 68,514 transcripts, where 98.95% of adapter-trimmed reads were aligned to the transcriptome assembly with minimap2[33] (Supplementary Method 3). We also aligned the assembled transcripts against the draft genome with minimap2 (Supplementary Table 10). A total of 66,866 (97.59%) assembled transcripts were aligned to the draft genome. Of these aligned transcripts, 21,423 (32.04%) transcripts have at least one split-alignments. Since split-alignments on a high-quality genomic reference typically indicate incorrectly assembled transcripts, we compared these split-alignments of assembled transcripts to STAR[34] alignments of the Illumina paired-end RNA-seq data against the draft genome. We found that 13,376 (62.44%) transcripts with split-alignments contain at least one split supported by at least one STAR alignment (Supplementary Method 7). This suggests that these transcripts were correctly assembled and the majority of split-alignments is likely a result of fragmented genic regions in the draft genome.

To understand the gene structure of transcripts contained in the genomic scaffolds, we supplied the RNA-Bloom2 assembly with hybrid error correction as full-length RNA sequences to PASA[35] to create a transcript structure annotation based on the draft genome. It is important to note that this annotation produced by PASA is only a partial representation of the Sitka spruce transcriptome due to fragmented genic regions. PASA generated an annotation consisting of 15,222 genes, 18,991 transcripts, 58,049 unique exons, 37,090 unique introns, and 19,079 poly(A) tails (Fig. 5a). There are more poly(A) tails than transcripts because PASA collapses transcripts with alternative polyadenylation. Overall, 95.7% of splice junctions from the PASA annotation overlaps with splice junctions in the Illumina paired-end RNA-seq data reported by STAR. We also tallied the frequencies of unique exons, introns, and transcripts per gene (Fig. 5b). On average, each gene has 3.8 exons, 2.4 introns, 1.2 transcripts. 59.12% genes contain 2 or more exons, and 16.1% genes contain at least 2 expressed transcripts. A maximum of 55 exons, 53 introns, 13 transcripts are observed per gene.

We calculated the length distributions of exons, introns, transcripts, genes, and poly(A) tails based on the output files from PASA (Fig. 5c). Exon lengths range from 10 to 18,115 nt with a primary peak at 116 nt and a slightly shorter secondary peak at 518 nt. Intron lengths

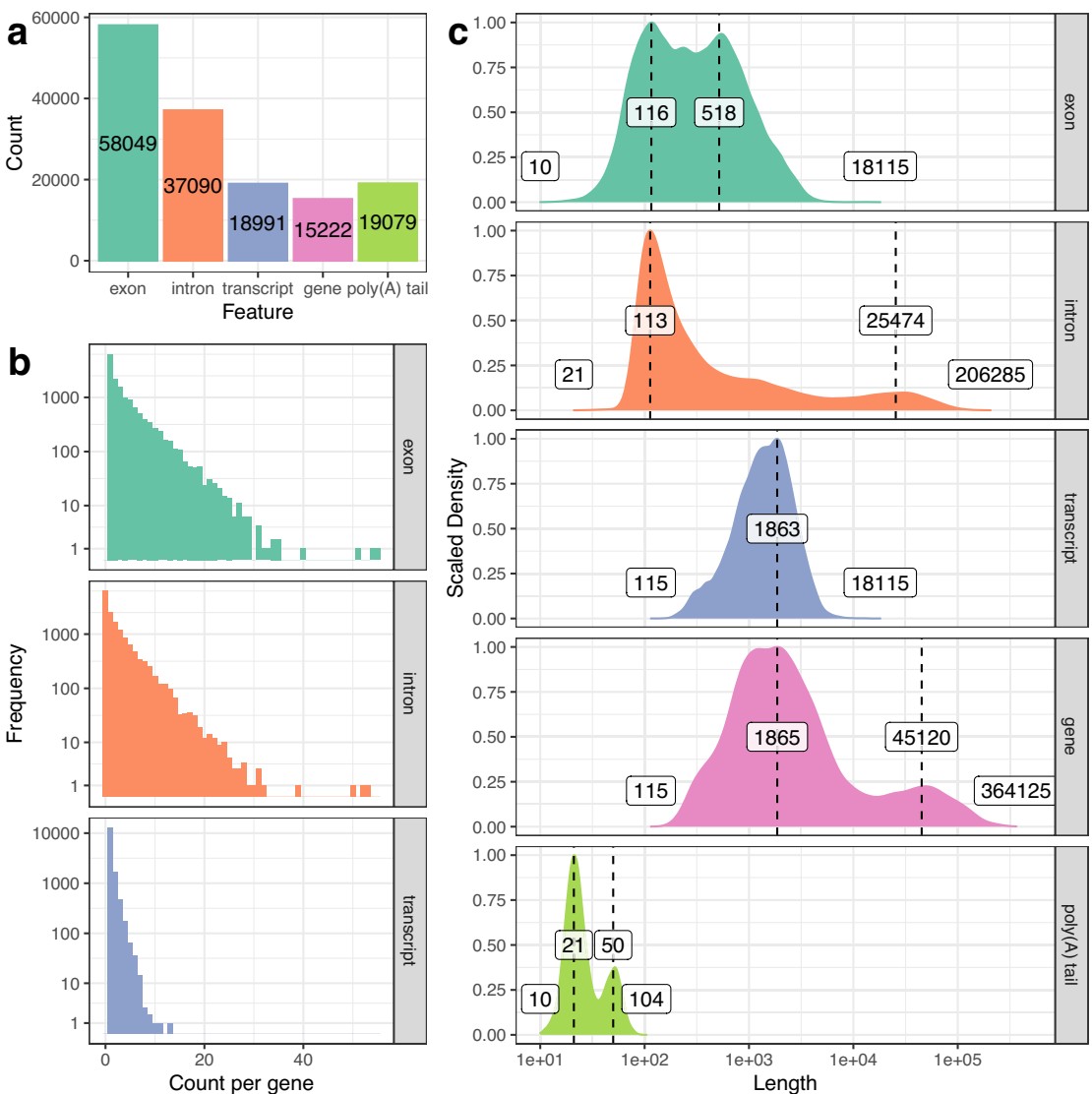

**Fig. 5 | Distributions of feature lengths and feature counts per gene for the Sitka spruce transcriptome. a** Total counts of exons, introns, transcripts, gene, and poly(A) tails. **b** The frequency of per-gene counts of exons, introns, and transcripts. The vertical "Frequency" axis is presented in logarithmic scale. **c** The length distributions of exons, introns, transcripts, genes, and poly(A) tails. The horizontal "Length" axis is presented in logarithmic scale. The vertical axis is scaled to the maximum value for each feature. The minimum and maximum values are indicated at both tails of the distributions. Peak values on the distributions are superimposed on the vertical dotted lines. Source data are provided as a Source Data file.

range from 21 to 206,268 nt with a primary peak at 113 nt and a much shorter secondary peak at 25,474 nt. 10.3% of introns are longer than 10,000 bp, which is in congruence with the long intron characteristic of conifers. Likely as a result of long introns, gene lengths range from 115 to 364,125 nt with a primary peak at 1865 nt and a secondary peak at 45,120 nt. Transcript lengths range from 115 to 18,115 nt with a peak at 1863 nt, which is nearly identical to the peak gene length. Poly(A) tail length ranges between 10 to 104 nt long with a primary peak at 21 nt and a shorter secondary peak at 50 nt. The bimodal poly(A) tail length distribution is also observed in *Arabidopsis* seeding transcriptomes[36].

We also investigated alternative transcript processing events in the assembled transcripts. PASA reports nine types of events (Fig. 6): spliced intron, retained intron, alternate acceptor, alternate donor, alternate exon, retain exon, skipped exon, starts in intron, and ends in intron. Spliced intron is the most common event (27.5%), followed by retained intron (24.3%). Retained intron and spliced intron are the most frequently co-occurring event types. Transcripts involving 3 or more event types are detected but are much rarer.

Finally, we applied the EnTAP pipeline[37] to produce protein sequence translation and functional annotation for the RNA-Bloom2 assembly (Supplementary Method 7). Using the functional annotation and similarity search to known spruce protein sequences, we have identified the following putative peptides: 15 terpene synthases (TPS), including 10 monoterpene synthases and seven diterpene synthases, 100 cytochrome P450 (CYP) peptides from 55 different subfamilies, and 17 NAM/ATAF/CUC (NAC) transcription factors from six different subfamilies. TPS, CYP, and NAC are gene families known for their contribution to constitutive and induced resistance to damage by the spruce weevils[31,38,39].

## Discussion

The rapid improvements to long-read sequencing technologies present a significant challenge to reference-free transcriptome assembly methods. As the throughput of long-read sequencers continues to increase, larger sequencing datasets are produced and thereby increasing the sequence assembly and analysis computational

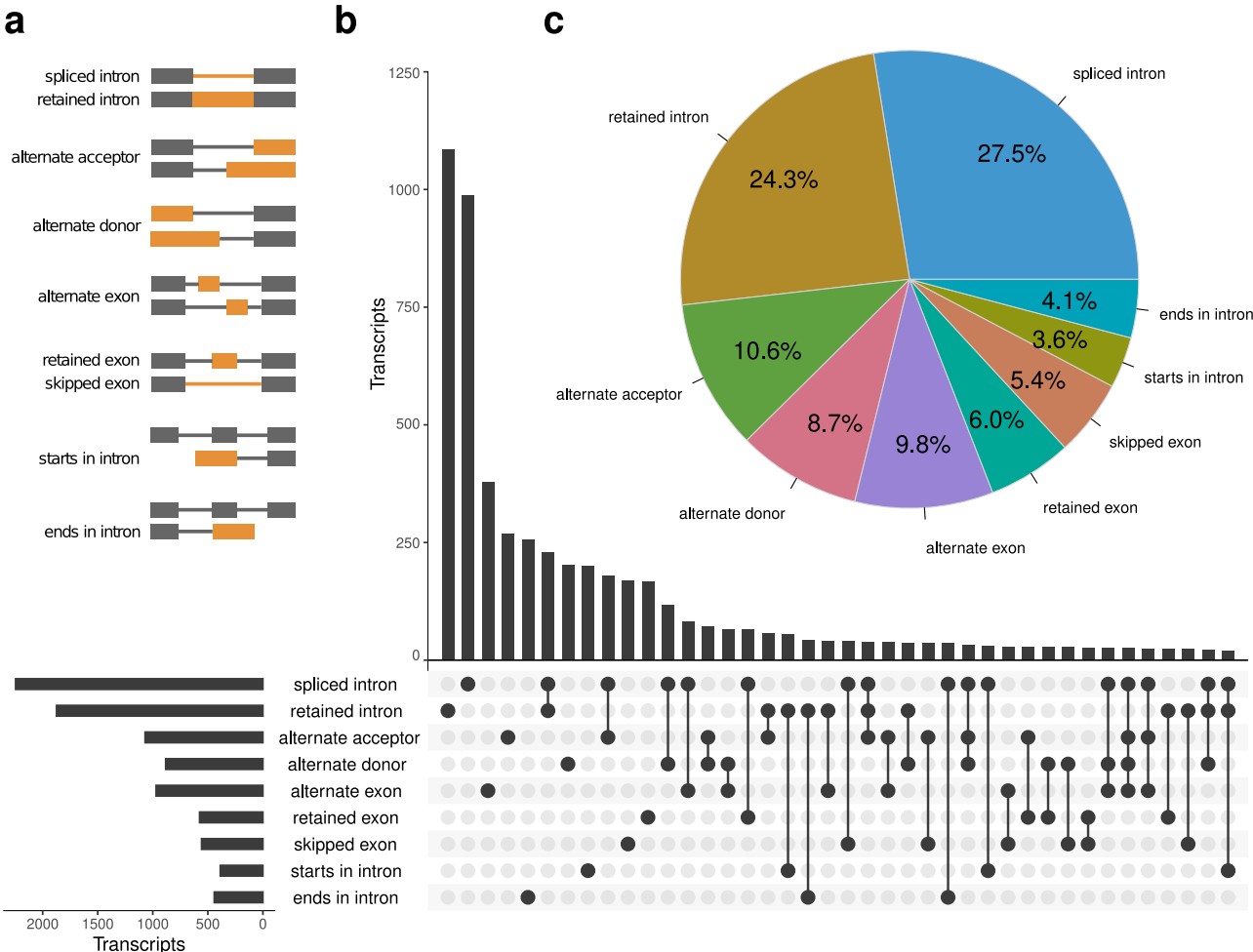

**Fig. 6 | Alternative transcript processing events in the Sitka spruce transcriptome. a** Nine types of alternative transcript processing events are presented as they are defined in PASA and depicted by connected grey and orange rectangles. The exons, introns, or splice-junctions involved in each event type are highlighted in orange. **b** The vertical bar chart in the UpSet plot shows the number of transcripts containing single event types and co-occurring event types, which are indicated by single dots and connected dots in the matrix, respectively. The horizontal bar chart in the UpSet plot shows the total number of transcripts containing each event type. **c** Relative proportions of all event types in the PASA annotation. Source data are provided as a Source Data file.

challenge. RNA-Bloom2 addresses this by digital normalization with strobemers. In our benchmarking with simulated data and spike-in control data, we showed that the computational performance and the assembly quality of RNA-Bloom2 significantly surpasses those of RATTLE, a previous reference-free transcriptome assembler that relies on the clustering of long reads. In particular, RATTLE has total wall-clock runtimes over nine times that of RNA-Bloom2. Therefore, digital normalization with strobemers, within the RNA-Bloom2 assembly workflow, was a successful application of the concept in the assembly of long-read sequencing data, and it was a superior alternative to clustering-based reference-free assembly. However, we note that digital normalization has somewhat detrimental effects on low-expressed isoforms as we have shown in our benchmarking experiment.

We note that reference-based assembly methods tend to run much faster than reference-free methods, but their overall assembly quality varies depending on the metric used and whether or not a transcriptome annotation is included. In simulated data, String-Tie2_GTF has better recall than reference-free methods, but StringTie2 has lower recall and higher FDR than reference-free methods. FLAIR has the lowest FDR in the simulated data, but StringTie2_GTF has the lowest FDR in the spike-in data. It is important to note that StringTie2 does not strictly require a reference annotation in addition to the

reference genome, but FLAIR requires both a reference annotation and the reference genome. Therefore, the application of FLAIR is mainly limited to the discovery of novel isoforms while StringTie2 only requires a good-quality reference genome.

With the experimental mouse data, annotation-guided methods tend to generate less novel not-in-catalog (as defined by SQANTI3) isoforms and novel non-canonical junctions compared to reference-free methods. This is expected because the reference annotation is used to filter or correct inaccurate splice junctions in read alignments. On the contrary, reference-free methods do not use the reference annotation and the reference genome; they do not have this a priori information to correct inaccurate splice junctions. Although this suggests that annotation-guided assembly may be more accurate than reference-free assembly, these novel isoforms and splice junctions remain to be verified. Nevertheless, RNA-Bloom2 and annotation-based methods recovered similar amounts of annotated genes. Compared to reference-free methods, all reference-based approaches have reconstructed substantially more novel genes and intergenic isoforms in ONT cDNA and PacBio CCS replicates. This suggests that reference-based methods may still produce false positives when the reference annotation does not contain sufficient information.

As good quality reference annotations are not always readily available, transcriptome assembly methods must manage the lack of

known transcription start and end sites, which are crucial in distinguishing the orientation of transcripts and discerning transcripts from antisense overlapping genes. Unlike direct RNA-sequencing, this is a major challenge for cDNA sequencing data, where the strand of reads cannot always be safely assumed. RNA-Bloom2 overcomes this problem by identifying potential poly(A) tail-containing reads. In addition, RNA-Bloom2 filters edges in its overlap graph based on read counts of each edge and its incident nodes, thus removing false overlaps between sequences. The positive effects of these solutions in RNA-Bloom2 are supported by the relatively low false discovery rates of RNA-Bloom2 in our benchmarking experiments for simulated data. When using the mouse data with replicates, we note that RNA-Bloom2 has reconstructed more FSM, ISM, NNC, and antisense isoforms than other methods. These metrics may be inflated by RNA-Bloom2's high redundancy – an aspect we observed when analyzing experimental but not simulated data. We, therefore, conclude that RNA-Bloom2 has room for improvement in recognizing noise and reducing the number of redundant transcripts in experimental data.

In our analyses of the Sitka spruce transcriptome, we illustrated that RNA-Bloom2 assemblies have higher BUSCO completeness than input reads and a RATTLE assembly. We note that a portion of our assembled transcripts have split-alignments across genome scaffolds, but the majority of them are supported by paired-end short reads. We expect that a transcript-informed targeted gene reconstruction[40], using a long-read reference-free transcriptome assembly by RNA-Bloom2, may significantly improve the discovery of new splice isoforms and the annotation of genes.

In summary, we illustrated the performance of RNA-Bloom2 with respect to state-of-the-art long-read transcriptome assembly methods, highlighting the strengths and weaknesses of each. We showed that RNA-Bloom2 is suitable for both ONT and PacBio sequencing technologies and it is competitive with reference-based methods. We expect RNA-Bloom2 to be scalable to increasing volumes of long-read data, and we anticipate RNA-Bloom2 will facilitate the gene annotation and transcriptome analyses of many species to be investigated.

## Methods

### Alignment-free error correction

We have modified the error correction routine for short reads in RNA-Bloom to support error correction in long reads. RNA-Bloom2 uses a $k$-mer size of 25 by default unless specified otherwise by the user. First, $k$-mers and their multiplicities in the input reads are stored in a Bloom filter de Bruijn graph, which uses the same memory-efficient data structure introduced in our previous work[19]. However, paired $k$-mers that were utilized in the short-read assembly algorithm are not used due to higher error rates of long reads. Reads are then split into fixed-length tiles (default: 500 nt) that are evaluated independently of each other. To account for varying transcript expression levels, a multiplicity threshold is dynamically determined (See Supplementary Fig. 4 for more details about this procedure) within each tile to identify "weak" and "solid" $k$-mers, which have multiplicities lower than or equal to the threshold, respectively. Weak $k$-mers represent potentially erroneous regions in the read while solid $k$-mers represent error-free regions in the read. To avoid introducing incorrect edits to the read, weak $k$-mers are replaced with an alternative path of solid $k$-mers in the de Bruijn graph only if this path shares a high sequence identity (default: 70%) as the target region spanned by the weak $k$-mers. The tiling nature of this routine ensures that more refined multiplicity thresholds are set for sub-regions in the read. The error correction process for each read may be repeated for additional iterations if at least one tile was modified in the previous iteration. In each successive iteration, the tiling positions are shifted by half a tile length to allow errors at tile boundaries in the previous iteration to be corrected. After error correction is completed for the read, $k$-mers

from all neighboring tiles are joined together and are assembled into an edited read sequence.

### Digital normalization with strobemers

Digital normalization is intended to reduce the overall read depth to a much lower target depth (default: 3) by identifying the minimal longest reads set (MLRS) supporting the target depth. Digital normalization reduces the computational resource requirements of subsequent stages, and it is most effective in reducing the number of reads for high-expressed transcripts, which are the main culprit for long runtimes and high memory usage in read-to-read alignments. The read depths represented by the MLRS are approximated by multiplicities of strobemers, which is an error-tolerant alternative to $k$-mers for sequence comparison. Three variants of strobemers have been introduced in previous work[18]: minstrobes, randstrobes, and hybridstrobes. In RNA-Bloom2, we used randstrobes of order three because it was shown to perform favorably on transcriptome data. Strobemer multiplicities are tracked by a counting Bloom filter, which is populated as reads are added to the MLRS.

Digital normalization begins by first sorting the input reads by their length in descending order. Only one read is evaluated at a time to maintain proper tracking of read depth in the MLRS. A read is designated as represented by the MLRS if nearly the entire read (default threshold of 50 nt from the read extremities) contains overlapping strobemers with multiplicities at or above the target depth (Supplementary Fig. 5). A read is designated as not represented by the MLRS if it has a region not containing any strobemers with multiplicities at or above the target depth. Each non-represented read is added to the MLRS and its strobemer multiplicities (that are lower than the target depth) would be incremented by one before the next read is evaluated. Represented reads are not included in the MLRS and their strobemer multiplicities are not incremented.

### Read trimming and splitting

RNA-Bloom2 relies on minimap2 for overlapping reads against each other to identify sufficiently covered regions of each read. By default, the minimum required read depth for long-read assembly is set to three in RNA-Bloom2; a sufficiently covered region of a read must overlap with at least two other reads. Insufficiently covered head and tail regions of the reads are trimmed. Reads containing insufficiently covered internal region(s) are potentially chimera artifacts. Therefore, reads are also split at these internal regions into shorter segments that are still sufficiently supported by other reads (See Supplementary Fig. 6 for more details). Completely contained reads are removed.

### Unitig assembly

Trimmed reads are overlapped against each other with minimap2 to construct an overlap graph where the vertices and edges are reads and their overlaps, respectively. As was done in the read trimming stage, contained reads are removed. If the input data is strand-specific (e.g. ONT dRNA), only alignments on the same strand are retained and the overlap graph would only contain vertices for the forward strand. If the input data is not strand-specific (e.g. ONT cDNA), then vertices for both strands are created in the overlap graph and the overlap graph is pruned based on whether each read contains poly(A) tail or poly(T) head (Supplementary Fig. 7). The overlap graph is simplified by removing transitive edges. Unitigs are derived by assembling reads along unambiguous paths in the overlap graph.

### Unitig polishing

Although alignment-free error correction has been performed on the reads that were used to generate unitigs, there are still residual base errors that can be polished using an alignment-based approach. Output reads from the error correction stage are aligned to the unitigs with minimap2. To avoid unintentional removal of short alternatively

spliced exons during polishing, only alignments with large indels (default: > 50 nt) or low sequence identity (default: <70%) are removed. The filtered alignments are passed to Racon[41] for polishing the unitigs.

## Transcript assembly

An overlap graph of polished unitigs is constructed based on minimap2 overlaps between polished unitigs. Reusing the read alignments from the unitig polishing stage, the overlap graph is annotated with: (i) length-normalized read counts for the unitigs, and (ii) the number of reads spanning across the unitig overlaps. The length-normalized read counts for unitigs are measured as the number of aligned bases divided by the total length of the unitig. The length normalization is primarily intended to account for reads that align partially to more than one unitig, where a higher count is attributed to the unitig with more aligned bases. Without length normalization, read counts tallied from reads aligned to unitig overlaps would be double-counted, which is particularly detrimental when reads align to false-overlaps between unitigs. Therefore, the normalized read counts provide a means to discern false overlaps between unitigs that belong to transcripts with different expression level magnitudes.

If the input data is not strand-specific, then the overlap graph is pruned as it was done in unitig assembly, and the read alignments are also examined for poly-A tail reads that are aligned to the unitigs. The unitigs are reoriented based on the poly-A tail read alignment orientations and the overlap graph is filtered accordingly (Supplementary Fig. 7). This procedure is crucial in discerning transcripts originating from overlapping genes on opposite strands of the chromosome. In addition, edges in the overlap graph are filtered by applying a binomial test on the number of reads supporting the edge with respect to the normalized read counts of the incident vertices (Supplementary Method 8).

After all filtering on the overlap graph has been performed, vertices are sorted by their read counts in descending order. Each vertex serves as the seed for a bidirectional greedy extension path with each extension choosing the neighbor vertex with the highest read count. Greedy extension terminates upon reaching either a dead-end, a cycle, or a vertex with a read count of zero. The reads along this path are assembled into a transcript. All vertices along this path would be flagged from seeding new extension paths, and their read counts are decremented by the minimum read count in the path. Transcript assembly is complete when all vertices have been visited.

## Benchmark dataset simulation

We used Trans-NanoSim v3.1.0 to simulate ONT cDNA and dRNA datasets based on the mouse annotation for GRCm39 from LRGASP (Synapse accession "syn25683629"). Two mouse samples from the LRGASP Consortium (accessions "ENCFF232YSU" and "ENCFF349BIN") were selected for training Trans-NanoSim sequencing profiles for cDNA and dRNA data, respectively. Sequencing adapters were trimmed from raw reads using Pychopper v2.5.0[21] (Supplementary Method 1). Since no adapters were detected in the dRNA data, the raw reads were supplied to Trans-NanoSim for training the dRNA profile. On the contrary, adapters were found in the cDNA data; the adapter-trimmed "full-length" and "rescued" reads, as defined by Pychopper, were supplied to Trans-NanoSim for training the cDNA profile. We discarded all simulated reads defined as "unaligned" by Trans-NanoSim and we subsample the "aligned" simulated reads to 2, 10, and 18 million reads using seqtk[42]. All software command parameters are documented in Supplementary Method 4.

## Spike-in control reads extraction

For the ONT cDNA samples, we only used the adapter-trimmed "full-length" and "rescued" reads, as defined by Pychopper. Using minimap2 2.24-r1122, reads from three replicates for each platform were aligned against the hybrid reference genome of mouse and spike-ins provided by LRGASP. Only reads that are aligned uniquely to ERCC and SIRV sequences are kept (Supplementary Method 5).

## Transcriptome assembly benchmarking

The command parameters for each assembler are documented in Supplementary Method 6. For the simulated datasets, transcriptome assemblies are aligned against the mouse reference transcriptome from LRGASP with minimap2. The output alignment PAF files are processed with our in-house Python script 'tns_eval.py', which is available at https://github.com/bcgsc/rnaseq_utils. Only alignment segments of at least 100 nt in length, at least 95% sequence identity, and indels smaller than 70 nt in length are considered. The ground truth transcript set is determined using the transcript identifiers in the simulated read names. Since not all known transcripts were simulated, the truth set is a subset of the annotation. Any transcripts that are not in the truth set are designated as false positives. If an assembled sequence aligns equally well with both a truth set transcript and a false-positive transcript, the assembled sequence would be assigned to the truth set instead of the false-positive. Any assembled sequences that have split alignments to more than one transcript are designated as misassemblies.

For the spike-in datasets, transcriptome assemblies are aligned against the ERCC and SIRV sequences with minimap2. Since the ground truth transcript set is identical to the spike-in annotated transcripts, there are no false-positives. However, misassemblies are still detected as it was done for the simulated datasets.

## Calculating isoform assembly precision

The isoform assembly precision metric is calculated based on SQANTI3's isoform classification. The command parameters for SQANTI3 are documented in Supplementary Method 11. SQANTI3 classifies assembly contigs into different categories according to how they match the reference transcript models. Full splice match (FSM) and incomplete splice match (ISM) are the only two categories that correspond to reference isoform with matching splice junctions. To determine isoform assembly precision for simulated data, the transcriptome annotation is filtered to include only isoforms that were simulated. Note that a total of six filtered transcriptome annotations were generated for the simulated data (two protocols, three read set sizes). When SQANTI3 is run with the filtered transcriptome annotation, any incorrect assemblies would be assigned to categories other than FSM or ISM. Therefore, isoform assembly precision is calculated as (FSM + ISM)/(FSM + ISM + other categories).

## Sitka spruce transcriptome analysis

All software command parameters are documented in Supplementary Method 7. The ONT cDNA reads were base called with Guppy v5.0.15. Since there are non-standard adapter and primer sequences, we used Porechop instead of Pychopper. We assembled the adapter-trimmed reads with RNA-Bloom2 v2.0.0 and short-read RNA-seq data (See Data Availability statement) from previous work[31] were also included only for error correction of the ONT reads. The transcriptome completeness was benchmarked with BUSCO v5.3.2[32] and the embryophyte core gene set (odb10). The resulting RNA-Bloom2 assembly was supplied to PASA v2.5.2[35] for gene structure annotation, using minimap2 for transcriptome alignments against the draft genome. The figure for alternative splicing was generated with UpSetR[43]. The transcriptome assembly was annotated with EnTAP v0.10.8-beta[37] using TransDecoder v5.3.0[44] for protein sequence translation. Functional annotation was assigned based on Swiss-prot plant proteins[45], UniRef90 gene clusters[46], embryophyte orthologs from OrthoDB10[47], and high-quality proteins derived from NCBI RefSeq 99[48]. We performed the annotation of TPS, CYP and NAC through a BLASTP search against target spruce protein sequences reported previously[31,38,39], with minimum match of 95% identity and 90% query coverage.

## Statistics and reproducibility

No biological sample collection was performed in this study. Therefore, no statistical method was used to predetermine sample size and there were no randomized experiments. Blinding of data is also not relevant. Sequencing reads were only excluded if they are deemed as poor quality by the adapter-trimming softwares. For reproducibility of analyses in this study, exact software commands are provided in Supplementary Information.

## Reporting summary

Further information on research design is available in the Nature Portfolio Reporting Summary linked to this article.

## Data availability

The source data for generating all figures in this study are provided in the Source Data file. The simulated data generated in this study has been deposited in Dryad at https://doi.org/10.5061/dryad.cc2fqz68w (ref. 49). The rebasecalled Nanopore sequencing data for the Sitka spruce cDNA sample has been deposited in the Sequence Read Archive (SRA) with run accession "SRR19510936". The GRCm39-based mouse reference genome from the LRGASP is available on Synapse with accession "syn25683365". The GENCODE VM27-based mouse transcriptome annotation set from the LRGASP is available on Synapse with accession "syn25683629". The sequencing data from the LRGASP are available in the ENCODE Project repository with accessions: "ENCFF349BIN", "ENCFF412NKJ", "ENCFF765AEC", "ENCFF232YSU", "ENCFF288PBL", "ENCFF683TBO", "ENCFF313VYZ", "ENCFF667VXS", "ENCFF874VSI", "ENCFF696TCH", and "ENCFF751FTE". The Illumina sequencing data for Sitka spruce from a previous study[31] is available on SRA with accessions: "SRR5949081", "SRR5949082", "SRR5949083", "SRR5949084", "SRR5949085", "SRR5949086", "SRR5949087", "SRR5949088", "SRR5949089", "SRR5949090", "SRR5949091", and "SRR5949092". All other relevant data supporting the key findings of this study are available within the article and its Supplementary Information files or from the corresponding author upon reasonable request. Source data are provided with this paper.

## Code availability

RNA-Bloom2 (v2.0.0) is implemented in Java, and it is publicly available under GPLv3 license on GitHub at https://github.com/bcgsc/RNA-Bloom (ref. 50). The scripts we wrote to produce and analyze our results are also publicly available on GitHub at https://github.com/bcgsc/rnaseq_utils and https://github.com/bcgsc/rnabloom2_manuscript.

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

## Acknowledgements

The authors would like to thank Dr. Steven Jones and Dr. Marco Marra at Canada's Michael Smith Genome Sciences Centre (GSC) for graciously providing their nanopore sequencing data to test an earlier version of RNA-Bloom2, Dr. Kieran O'Neil and Vanessa Porter at GSC for evaluating the performance of an earlier version of RNA-Bloom2, Dr. Justin Whitehill at North Carolina State University and Mack Yuen at Dr. Joerg Bohlmann's Lab for providing the sequences and annotations of Sitka spruce short-read transcriptome assemblies, Young Cheng at GSC for submitting the Sitka spruce nanopore sequencing data to the SRA, and the organizers of the LRGASP Consortium for making their sequencing data publicly available and sharing their evaluation results on an earlier version of RNA-Bloom2. This work was supported by Genome Canada and Genome British Columbia (grant number 243FOR to all authors); the National Institutes of Health (grant number 2R01HG007182-04A1 to all authors); the Natural Sciences and Engineering Research Council of Canada (NSERC); and the Canadian Institutes of Health Research (CIHR). The content of this work is solely the responsibility of the authors and does not necessarily represent the official views of the National Institutes of Health or other funding organizations.

## Author contributions

K.M.N. and I.B. conceived the study. K.M.N. designed and developed RNA-Bloom2 under the supervision of I.B. K.M.N., S.H., and C.Y. generated the simulated datasets and analyzed the LRGASP Consortium dataset. S.H., R.C., R.L.W., and I.B. advised on the benchmarking design. K.M.N., K.K.G., and R.C. analyzed the Sitka spruce dataset. All authors contributed to the interpretation of the results. KMN took the lead in writing the manuscript with input from all authors.

## Competing interests

The authors declare no competing interests.
