## [Peer Review File · Nature Communications]

REVIEWER COMMENTS

Reviewer #1 (Remarks to the Author):

The manuscript "Reference-free assembly of long-read transcriptome sequencing data with RNA-Bloom2" by Ka Ming Nip et al. describes development and application of a de novo transcriptome assembler RNA-Bloom2 devised for transcriptome long read sequence assembly. The authors demonstrate RNA-Bloom2 on simulated data and real data, compare its performance to alternative genome-based and genome-free methods, and finally demonstrate its efficacy in assembling the transcriptome for Sitka spruce, a plant that has a low-quality draft genome sequence available.

RNA-Bloom2 is the second method to be published that tackles this challenge of genome-free long-read transcriptome assembly, with the earlier published method being 'RATTLE'. The authors demonstrate RNA-Bloom2 to substantially outperform RATTLE and hence an important advance in this realm.

The manuscript is clearly well written, the methods well documented in the main and supplemental text, and figures are clearly illustrated. I appreciate the authors attention to detail and the overall high quality of the work as presented. I have only minor comments and suggestions for further revision as follows.

In the introduction, it's described (starting on line 87) "Sequence clustering-based assembly follows the divide-and-conquer paradigm and thus it requires less resources than methods that align all reads against each other. " and that RATTLE uses this. I was under the impression that RNA-Bloom2 would leverage some similar strategy, but it appears that it does not, and performs an all-vs-all comparison at the assembly stage. It would be useful to clarify why this is mentioned if it's not taken advantage of within the RNA-Bloom2 implementation.

The evaluation of the efficacy of digital normalization (starting on line 194) relies on stats regarding the fraction of reads that align to the reference genome. It would be more informative to instead examine the number of genes that are represented by those alignments. The ideal situation is that digital normalization reduces the number of reads, but the number of genomic loci represented remains largely the same.

For reporting on the benchmarking accuracy, it would be useful to have a combined measure of recall and precision such as an F1 metric. This would make it easier to examine the overall accuracy differences between methods. Also, in the case of false positives (eg. Figure 3C), it would be helpful to understand where these are being derived from. In the case of simulated data, the corresponding transcripts from which the reads are derived are known, and so are the false positive reconstructions mostly coming from mis-mappings or paralog alignments in the case of Stringtie2?

Alternative splicing isoform assembly accuracy is one of the primary challenges in long read transcriptome assembly, and including a section that directly addresses this aspect with the simulated data would be particularly useful, highlighting the differences in accuracy between the alternative reconstruction methods accordingly.

Line 596 – the binomial test is mentioned as a way of defining edges worth pruning. Could additional info be added regarding this, either in the main or supplemental text?

Minor text adjustments:

Line 35 of abstract states ", it sets up the groundwork ... " I'd suggest stating ', it further sets up ...' since RNA-Bloom2 is not the first development in this realm.

Line 216 states “gold standard” – I think ‘baseline’ might be more appropriate.

Reviewer #2 (Remarks to the Author):

The manuscript presents RNA-Bloom2, a long read assembler for RNA data. This manuscript fills a gap as it proposes a de novo assembly methods for RNA long reads, a kind of contribution the current literature lacks of. This method also deals with Nanopore reads, while the majority of contributions has focused on PacBio. Nanopore proposes to sequence direct RNA, which has numerous advantages but is known as more difficult because it combines more error and less depth. RNA-Bloom2 performs well on this data, it seems like a major asset. The authors previously published RNA- Bloom, an assembly method for RNA which is distinct from this one as it takes single cell RNA as an input. The manuscript shows competitive and qualitative assembly results on different datasets. I found the manuscript sufficiently well written to comprehend the work.

The method differs from the single other available de novo assemblers for RNA long reads. It cannot always compete with reference-based assemblers, which is perfectly fine and expected since they include more information from the start.

A first improvement I suggest is to clarify why “assembly” is still needed with these long reads. Many of them are expected to span, if not all, a very large part of the RNA molecule. Therefore, other approaches, with dropped the assembly part, identify reads belonging to each different transcripts and correct them, such as IsOnCorrect by Sahlin et al. This approach deserves to be compared to RNA-Bloom2.

The manuscript claims that the method is time and memory efficient, notably due to an in silico normalization, which seems supported by the experiments.

The manuscript contains a full description of the assembly metrics that are used to assess the work. It sums up the quality of the assembly on simulated data, complete and missing transcripts. It also details the effect of the expression on the results. Here I feel a clearer message on how well the method performs on low-expression transcripts would be helpful. Misassemblies and other errors are also reviewed.

When it comes to the other methods included in the benchmark, aside from what I already said about IsOnCorrect, I don't understand why StringTie2 is not shown using an annotation. FLAIR is described as the type of work relying on annotations, with the pros and cons it possibly brings, but I think this is a standard use for StringTie2 as well. This should strengthen the manuscript's claims concerning using annotations versus de novo.

The manuscript used simulated datasets which allow to assess results with a “golden truth” but have the disadvantage to not fully represent the difficulty of real dataset. It used spike-in data which is used in RNA-seq to control whether known expressed genes are taken into account by the method. The software is tested on eukaryotes (plants and animals). There is a correct number of reads and the reported error rates are in the expected range.

In summary, RNA-Bloom2's methodology relies on a read correction step based on k-mers, then the data undergoes digital normalization using strobemers, reads are trimmed and all versus all alignment is performed with minimap2 in order to build an overlap graph. So-called unitigs are extracted from the assembly graph, and then polished using the corrected reads. A second overlap graph is built using the polished data, from which the final contigs are produced.

RNA-Bloom2 uses different thirdparty tools, such as Minimap2, Trans-NanoSim or RACON, which occur to me as up-to-date methodology. Commands and software versions are reported in supplementary data, which is important towards reproducibility.

RNA-Bloom2 also makes a good use of ideas introduced or re-introduced in the context of long reads,

such as strobemers or the overlap graph for assembly.

There is room for improvement in the method section. I was convinced by the digital normalization approach, which I think is a good addition to the field's toolbox. There are other points I'd like to raise concerning the methods though.

-In p18 l511, a "Bloom filter de Bruijn graph" is not defined or at least pointing to some helpful reference. It can be a source of confusion for users.

-The text is sometimes not precise enough, for instance "and thus are split in shorted sub-sequences" p19 l558: what is shorter? Also what is the k value? I believe this should appear in the main text.

- RNA-Bloom2 includes a read correction step. Why does it seem that corrected reads used only for polishing, and not directly fed to the assembly pipeline?

Suggested minor improvements

- From p6: in the input reads that could not align to the final assembly after digital normalization, which proportion has been cast away during digital normalization? Does that tell us something about a read category that could be poorly treated (even though it will be no more than 3-5%, we know that rare events are really looked for too by RNA people)?

- In Figure 6, if I'm not mistaken, alternative start/end of transcripts are not alternative splicing events per se. They still account in the final variability, but they occur at a different step during RNA processing in the cell.

- I believe that publishing in a high impact journal means having high standard on the reproducibility, including the bioinformatics side. Nowadays, exemplary works dedicate a webpage (for instance on their Github) with all the instruction, links to data and material and commands to reproduce the experiments of the manuscript. I really feel that we should expect this more and more from methods papers. I am aware that it represents a lot of work, which is why I put this remark in the "minor improvement" section.

Reviewer #3 (Remarks to the Author):

This paper by Ming Nip et al. presents RNA-Bloom2, a reference-free transcript assembler for long reads data. This is an adaptation of a previous methodology from the same group to deal with longer and noisier reads. The authors compare their method to RATTLE, another reference-free transcriptome assembler, and StringTie2 and FLAIR, algorithms that use the reference annotation to guide transcript reconstruction. To evaluate the performance of the method they simulate data using Trans-NanoSim and take SRIVs reads generated by the LRGASP competition -a large community project to evaluate long reads transcriptome reconstruction methods, where RNA-Bloom participates-. They show a good performance of RNA-Bloom2, especially when compared to RATTLE. Finally, they apply the method to the reconstruction of the Stika spruce transcriptome, which is characterized. They find over 66,000 transcripts, the great majority of the genes expressing only one transcript.

As very well stated in the introduction, long reads are increasingly being used for the characterization of transcriptomes. Importantly, many papers report large expansions in the number of expressed and novel transcripts using these technologies. Additionally, long reads are used in cases where genome annotation is poor, representing an opportunity to characterize transcriptomes for non-model species. Since both Pacbio and Nanopore may produce more sequencing errors than Illumina, and their accompanying library preparation may introduce artifacts, the development of methods that can accurately identify transcripts from these data is extremely important. Therefore, RNA-Bloom2 is a relevant and timely contribution.

Since the method participated in the LRGASP competition and the paper uses data generated by this consortium, it was straightforward to compare the work in this manuscript with LRGASP. The LRGASP

methodology has been published as Registered Report in Nature Methods (https://springernature.figshare.com/registered-reports_nmmethods) and preliminary results were made public at the LRGASP site (<https://www.gencodegenes.org/pages/LRGASP/>). In particular, the goal of this paper matches Challenge 3 of LRGASP, which evaluates the scenario of transcriptome reconstruction without the utilization of a reference transcriptome annotation. The manuscript by Ming Nip complements the LRGASP effort as they use simulated datasets to evaluate the performance of the comparing methods, include RATTLE in the comparison, and they assess RNA-Bloom2 with PacBio data. None of these evaluation venues were part of LRGASP. On the contrary, the evaluation using SIRVs is redundant with LRGASP and can be compared.

Interestingly, when comparing the assessment provided in this paper to LRGASP results, there are important discrepancies, which are mostly related to how evaluation metrics are defined and how results are presented. I believe that the evaluation metrics presented in this paper are incomplete and do not provide a good assessment of the performance of the methods. Specifically, I have strong concerns as to how these metrics are capable of evaluating the false transcript calls, which is the biggest problem in long reads.

For example, the metric "complete reconstruction" is described as the transcripts reconstructed with at least 95% of their length. In this metric, there is no consideration of the conservation of the splicing pattern of the true transcript. If the allowed 5% difference would affect splice junctions, transcripts could have different splice patterns, be annotated as different transcripts in a reference transcriptome, and even result in different coded proteins. The current standard in the field is to evaluate the identity of the transcript models -at least when a reference genome is available for evaluation, such as the case here- by evaluating the splice junction pattern, frequently following the naming provided by SQANTI (PMID: 29440222) and possibly also the accuracy at 3' and 5' ends. A great majority of the novel transcripts claimed in recent long reads transcriptomics papers are "novel in catalog" and "novel not in catalog transcripts", according to the SQANTI nomenclature, which means they have small junction differences with respect to the reference. It is not clear to me if these changes would be considered a "complete reconstruction" by the definition proposed by the authors of this work, which make it difficult to evaluate the significance of the contribution with respect to the current literature. Therefore, authors should include a direct assessment of the accuracy at calling the correct combination of splice sites when using simulated and SIRVs data.

Also, the definition of false discovery rate (FDR) is intriguing, and I am not sure to which extent this is compensated by the misassembly metric. False positives were defined in this work as reference transcripts identified by the method but not present in the ground truth. This type of mistake -calling the wrong reference transcript- is unlikely to happen, especially if the assembly is not reference-guided. As I mentioned before, an important number of false detections are poor solutions to errors resulting in novel transcripts with small junction differences. However, the provided definition of misassembly -as having segments (how big?) from other transcripts- suggests they are looking for chimeric transcripts, which are not as frequent, and not for these small variations. Such small variations are, for example, detected in the Stika spruce results as alternative donors and acceptors, and account for 20% of the data. Would false alternative donors and acceptors be classified as misassembly? The definition of false calls in this paper needs to be better explained and possibly, expanded.

Since this paper evaluates RNA-Bloom and other methods using the LRGASP SIRVs dataset, results can be compared to those made public by the consortium for the ONT cDNA dataset in Challenge 3. The definition of recall for complete and partial transcripts are similar to the Sensitivity and Positive Detection Rates (PDR) in LRGASP. Interestingly values are similar for RNA-Bloom: ~60% Sensitivity and ~78% PDR to the complete and partial recall values of this manuscript. However, the false positive aspect is reflected very differently. While here a 9% Misassembly is reported, LRGASP indicates (on the same ONT cDNA SIRVs data), a 75% of FDR and a 0.25 Precision. This discrepancy needs to be explained.

There is also no data on the number of inferred transcripts by the methods in comparison to the simulated and SIRV transcripts and in how many cases the same reference transcript is detected by more than one assembled transcript. This is also a common source of extra-transcripts when collapsing is not complete. The Recall metric does not measure this source of redundancy. The values of the total number of detected transcripts by each method and how they related to the true transcripts should be provided.

Finally, both the simulated data and the SIRVs data have limitations as benchmarking tools. Simulated data may reproduce sequencing errors and partial degradation patterns but this is not the only source of error in the technology. The library preparation method also contributes to creating false molecules to be sequenced. For cDNA this includes polyA intrapriming and RT switching, among others, and for dRNA gaps in the Nanopore signal may result in false (partial) exon skipping. To which extent is this properly accounted for in the simulated that is unclear. On the other hand SIRVs represent simplified datasets, that cannot capture the complexity of a real sample. I would strongly recommend including in their evaluation replicated data of a real sample with a good reference annotation, and evaluating the types of detections (in the reference or not) and the consistency of calls across these replicates.

Other comments on the algorithm and remarks

Some aspects of the algorithm are unclear to me. How does digital normalization work? Will this normalization or other steps in the pipeline introduce any correction of the result? Does the method introduce any sequence correction based on long reads data?

One aspect of the long reads data we have observed is that tools report a high number of transcripts with one or few supporting reads. The RNA-bloom2 method evaluates the multiplicity of k-mers to identify strong and weak ones. Could you indicate which is the distribution of these multiplicity values and where the threshold is decided in each case? Supplementary figure 3 shows the concepts but not the actual values.

When evaluating simulated and SIRVs data, How many total transcripts were detected? what is the distribution of transcripts per gene? Does the distribution match the true transcripts? Also, results should be broken down by accurate transcripts from multi-transcript genes and accurate transcripts from single transcript genes.

What is the percentage of non-canonical splice sites present in the RNA-Bloom2 results?

RESPONSE TO REVIEWERS' COMMENTS

Reviewer 1's remarks

“In the introduction, it’s described (starting on line 87) “Sequence clustering-based assembly follows the divide-and-conquer paradigm and thus it requires less resources than methods that align all reads against each other. “ and that RATTLE uses this. I was under the impression that RNA-Bloom2 would leverage some similar strategy, but it appears that it does not, and performs an all-vs-all comparison at the assembly stage. It would be useful to clarify why this is mentioned if it’s not taken advantage of within the RNA-Bloom2 implementation.”

Response:

We have clarified this in the Results section. RNA-Bloom2 only performs all-vs-all comparisons after digital normalization. Since the number of sequences retained after digital normalization is expected to be much lower than the number of raw input reads, sequence clustering is not necessary in RNA-Bloom2. As shown in Figure 2, RNA-Bloom2 (with no sequence clustering) still substantially outperforms RATTLE in computational performance.

“The evaluation of the efficacy of digital normalization (starting on line 194) relies on stats regarding the fraction of reads that align to the reference genome. It would be more informative to instead examine the number of genes that are represented by those alignments. The ideal situation is that digital normalization reduces the number of reads, but the number of genomic loci represented remains largely the same.”

Response:

We thank the reviewer for the suggestion. We have analyzed the effect of digital normalization on the number of genes represented using experimental data for ONT dRNA, ONT cDNA, and PacBio CCS. We have incorporated our results in the Results section (sub-section “Assembly benchmarking with simulated datasets”), Supplementary Table 4, and Supplementary Figure 15. Overall, the number of genes represented in the digitally normalized reads is 95.6-98.3% of those in the raw reads. The genes discarded are predominantly lowly expressed. Our results agree with the reviewer’s expectation.

“For reporting on the benchmarking accuracy, it would be useful to have a combined measure of recall and precision such as an F1 metric. This would make it easier to examine the overall accuracy differences between methods. Also, in the case of false positives (eg. Figure 3C), it would be helpful to understand where these are being derived from. In the case of simulated data, the corresponding transcripts from which the reads are derived are known, and so are the false positive reconstructions mostly coming from mis-mappings or paralog alignments in the case of Stringtie2?”

Response:

We have included F1 scores for Figures 3C and 4C. In response to Reviewer 3's remarks, we have incorporated false-positive reference transcripts, misassemblies, large indel contigs, and unclassified contigs (see Table 1) into the calculation for false-discovery rates (see Figures 3B and 4B). False positive reference transcripts are alternative isoforms of the simulated isoforms. We speculate that false-positive reference transcripts are the result of incorrect assembly in multi-isoform genes instead of incorrect mappings. This is not limited to StringTie2.

“Alternative splicing isoform assembly accuracy is one of the primary challenges in long read transcriptome assembly, and including a section that directly addresses this aspect with the simulated data would be particularly useful, highlighting the differences in accuracy between the alternative reconstruction methods accordingly.”

Response:

We have calculated the isoform assembly precision based on the isoform classification from SQANTI3, as suggested by Reviewer 3. We have incorporated our findings in the Results section, and we have added Supplementary Figure 8 in support.

“Line 596 – the binomial test is mentioned as a way of defining edges worth pruning. Could additional info be added regarding this, either in the main or supplemental text?”

Response:

We have added Supplementary Method 8 to clarify this.

“Line 35 of abstract states “, it sets up the groundwork ... “ I'd suggest stating ‘, it further sets up ...’ since RNA-Bloom2 is not the first development in this realm.

Line 216 states “gold standard” – I think ‘baseline’ might be more appropriate.”

Response:

Thanks for these suggestions. We have corrected these accordingly.

Reviewer 2's remarks

“A first improvement I suggest is to clarify why “assembly” is still needed with these long reads. Many of them are expected to span, if not all, a very large part of the RNA molecule. Therefore, other approaches, with dropped the assembly part, identify reads belonging to each different transcripts and correct them, such as IsOnCorrect by Sahlin et al. This approach deserves to be compared to RNA-Bloom2.”

Response:

We provided additional results in the Results section and added Supplementary Figure 1. Our results indicate that RNA-Bloom2 assemblies offer more complete transcripts and less false discoveries compared to isONcorrect. Therefore, there is indeed a need for assembly of long read transcriptomic data.

“The manuscript contains a full description of the assembly metrics that are used to assess the work. It sums up the quality of the assembly on simulated data, complete and missing transcripts. It also details the effect of the expression on the results. Here I feel a clearer message on how well the method performs on low-expression transcripts would be helpful.”

Response:

We have clarified this message in the manuscript on page 11 and our results for Supplementary Figures 2 and 3.

“When it comes to the other methods included in the benchmark, aside from what I already said about IsOnCorrect, I don’t understand why StringTie2 is not shown using an annotation. FLAIR is described as the type of work relying on annotations, with the pros and cons it possibly brings, but I think this is a standard use for StringTie2 as well. This should strengthen the manuscript’s claims concerning using annotations versus de novo.”

Response:

We have included results for StringTie2 guided by the annotation (denoted as “StringTie2_GTF” throughout the manuscript). Figures 2 and 3, Supplementary Tables 6 and 7, Supplementary Figures 2 and 3, and Supplementary Method 6 have been updated as a result.

“-In p18 l511, a “Bloom filter de Bruijn graph” is not defined or at least pointing to some helpful reference. It can be a source of confusion for users.”

Response:

We have clarified this in the Methods section (sub-section “Alignment-free error correction”) and added a reference to our previous publication, which describes how it was constructed.

“-The text is sometimes not precise enough, for instance “and thus are split in shorted sub-sequences” p19 l558: what is shorter? Also what is the k value? I believe this should appear in the main text.”

Response:

The shorter sub-sequence refers to segments (in a potentially chimeric read) that are still sufficiently supported by other reads.

RNA-Bloom2 uses a k -mer size of 25 by default unless specified otherwise by the user.

We have clarified these details in the Methods section.

“- RNA-Bloom2 includes a read correction step. Why does it seem that corrected reads used only for polishing, and not directly fed to the assembly pipeline?”

Response:

We think our reviewer is not interpreting the pipeline description correctly. As shown in Figure 1, Stage 1 (Error Correction) generates “corrected reads” that has two out-going arrows to Stages 2 (Digital Normalization) and 5 (Polishing). Stage 2 leads the corrected sequences to Unitig Assembly by taking them through Digital Normalization and Trimming & Splitting. Therefore, a subset of corrected reads is fed into the subsequent assembly stages and all corrected reads are used again during Polishing.

“- From p6: in the input reads that could not align to the final assembly after digital normalization, which proportion has been cast away during digital normalization? Does that tell us something about a read category that could be poorly treated (even though it will be no more than 3-5%, we know that rare events are really looked for too by RNA people)?”

Response:

In a perfect scenario, all unaligned reads should be discarded, and a large portion of aligned reads should be discarded without lowering any number of genes represented by the reads. In practice, both unaligned and aligned reads can be discarded during error correction and digital normalization. As also suggested by Reviewer 1, we have provided additional results for the number of genes represented by raw reads, corrected reads, and digitally normalized reads. Please see our new results in the Results section (sub-section “Alignment-free error correction and digital normalization demonstrates utility in multiple data types”).

“- In Figure 6, if I’m not mistaken, alternative start/end of transcripts are not alternative splicing events per se. They still account in the final variability, but they occur at at different step during RNA processing in the cell.”

Response:

We thank the reviewer for catching this misnomer. We have renamed it to “alternative transcript processing events” in the main text and Figure 6.

“- I believe that publishing in a high impact journal means having high standard on the reproducibility, including the bioinformatics side. Nowadays, exemplary works dedicate a webpage (for instance on their Github) with all the instruction, links to data and material and commands to reproduce the experiments of the manuscript. I really feel that we should

expect this more and more from methods papers. I am aware that it represents a lot of work, which is why I put this remark in the “minor improvement” section.”

Response:

We have included a dedicated Github repository for detailing the bioinformatics analyses performed in our study and reported in the manuscript.

Reviewer 3’s remarks

“Interestingly, when comparing the assessment provided in this paper to LRGASP results, there are important discrepancies, which are mostly related to how evaluation metrics are defined and how results are presented. I believe that the evaluation metrics presented in this paper are incomplete and do not provide a good assessment of the performance of the methods. Specifically, I have strong concerns as to how these metrics are capable of evaluating the false transcript calls, which is the biggest problem in long reads.”

Response:

The assemblies we provided to the LRGASP organizers were from more than one year ago and they were generated using an older version of RNA-Bloom (v1.4.3) that has a less sophisticated algorithm for long-read assembly, and the version information is duly reported in both manuscripts. Most importantly, the input reads that we used at the time include the “unclassified” reads (as defined by Pychopper), which are a significant source of noise as we had discovered after the fact. In this manuscript, we are only using the “fulllength” and “rescued” reads (as defined by Pychopper). Based on the valuable preliminary assessment results provided by the LRGASP organizers, we have since made significant improvements to our algorithm, which resulted in RNA-Bloom2. Therefore, a level of discrepancies between our results and those presented by the LRGASP organizers should be expected. However, we highly appreciate the reviewer’s concerns and suggestions. We have since modified the definition of our evaluation metrics and provided additional analyses suggested by all three reviewers.

“For example, the metric “complete reconstruction” is described as the transcripts reconstructed with at least 95% of their length. In this metric, there is no consideration of the conservation of the splicing pattern of the true transcript. If the allowed 5% difference would affect splice junctions, transcripts could have different splice patterns, be annotated as different transcripts in a reference transcriptome, and even result in different coded proteins. The current standard in the field is to evaluate the identity of the transcript models -at least when a reference genome is available for evaluation, such as the case here- by evaluating the splice junction pattern, frequently following the naming provided by SQANTI (PMID: 29440222) and possibly also the accuracy at 3’ and 5’ ends. A great majority of the novel transcripts claimed in recent long reads transcriptomics papers are “novel in catalog” and “novel not in catalog transcripts”, according to the SQANTI nomenclature, which means they have small junction differences with respect to the reference. It is not clear to me if these changes would be considered a “complete

reconstruction” by the definition proposed by the authors of this work, which make it difficult to evaluate the significance of the contribution with respect to the current literature. Therefore, authors should include a direct assessment of the accuracy at calling the correct combination of splice sites when using simulated and SIRVs data.”

Response:

We thank our reviewer for the concern raised and for the opportunity to clarify our logic. A contig that contributes to partial/complete reconstruction must have only a single-block alignment (at least 100 nt in length, 95% sequence identity with indels no larger than 70 nt) against the corresponding true-positive transcript. If another segment of this contig aligns (with the same criteria mentioned previously) to a different reference isoform, then this contig would be classified as a misassembly instead of partial/complete reconstruction. In addition, contigs with incorrect splice junctions (e.g. due to alternative donors/acceptors) should result in alignments with large indels against the reference transcripts, and therefore these contigs are designated as false discoveries instead of complete/partial reconstruction. Therefore, incorrectly assembled splice variants of true-positive transcripts are accounted for. Nevertheless, we understand the reviewer’s concerns and we have provided SQANTI3 results to complement our analysis results.

“Also, the definition of false discovery rate (FDR) is intriguing, and I am not sure to which extent this is compensated by the misassembly metric. False positives were defined in this work as reference transcripts identified by the method but not present in the ground truth. This type of mistake -calling the wrong reference transcript- is unlikely to happen, especially if the assembly is not reference-guided. As I mentioned before, an important number of false detections are poor solutions to errors resulting in novel transcripts with small junction differences. However, the provided definition of misassembly -as having segments (how big?) from other transcripts- suggests they are looking for chimeric transcripts, which are not as frequent, and not for these small variations. Such small variations are, for example, detected in the Stika spruce results as alternative donors and acceptors, and account for 20% of the data. Would false alternative donors and acceptors be classified as misassembly? The definition of false calls in this paper needs to be better explained and possibly, expanded.”

Response:

We thank our reviewer for this valuable remark. We have modified our definition of false discoveries and the calculation of FDR. False discoveries now include (i) false-positive reference transcripts, (ii) intragene and intergene misassemblies, (iii) large indel contigs, (iv) unclassified contigs. Please see our updated Table 1 for more details about these metrics. Figures 3 and 4 have also been updated as a result. As mentioned in our response to the previous point, segments of at least 100-nt in length are considered and false alternative donors/acceptors are also accounted for.

“Since this paper evaluates RNA-Bloom and other methods using the LRGASP SIRVs dataset, results can be compared to those made public by the consortium for the ONT

cDNA dataset in Challenge 3. The definition of recall for complete and partial transcripts are similar to the Sensitivity and Positive Detection Rates (PDR) in LRGASP. Interestingly values are similar for RNA-Bloom: ~60% Sensitivity and ~78% PDR to the complete and partial recall values of this manuscript. However, the false positive aspect is reflected very differently. While here a 9% Misassembly is reported, LRGASP indicates (on the same ONT cDNA SIRVs data), a 75% of FDR and a 0.25 Precision. This discrepancy needs to be explained.”

Response:

A level of discrepancy is expected, as stated above. Since we extracted reads that aligned uniquely to the SIRV references to create our dataset for benchmarking (as shown in Supplementary Method 5), some reads that may correspond to noise may have been removed as a result. In addition, we have adjusted our assessment metrics, such as FDR, and updated our results. Our updated results show that RNA-Bloom2 does have higher FDR than we had previously presented in our initial submission.

“There is also no data on the number of inferred transcripts by the methods in comparison to the simulated and SIRV transcripts and in how many cases the same reference transcript is detected by more than one assembled transcript. This is also a common source of extra-transcripts when collapsing is not complete. The Recall metric does not measure this source of redundancy. The values of the total number of detected transcripts by each method and how they related to the true transcripts should be provided.”

Response:

We have provided a new metric for assembly redundancy, which is defined as the ratio between (i) the number of contigs representing true-positive transcripts with complete or partial reconstruction and (ii) the number of true-positive transcripts with complete or partial reconstruction. Therefore, a value higher than 1 indicates redundancy. We have added results for this metric in Figures 3D and 4D.

“Finally, both the simulated data and the SIRVs data have limitations as benchmarking tools. Simulated data may reproduce sequencing errors and partial degradation patterns but this is not the only source of error in the technology. The library preparation method also contributes to creating false molecules to be sequenced. For cDNA this includes polyA intrapriming and RT switching, among others, and for dRNA gaps in the Nanopore signal may result in false (partial) exon skipping. To which extent is this properly accounted for in the simulated that is unclear. On the other hand SIRVs represent simplified datasets, that cannot capture the complexity of a real sample. I would strongly recommend including in their evaluation replicated data of a real sample with a good reference annotation, and evaluating the types of detections (in the reference or not) and the consistency of calls across these replicates.”

Response:

Although simulated data and spike-in control data may have their own limitations, all assembly methods evaluated in our benchmarking analyses are affected equally by these limitations. Therefore, our benchmarking analyses on these datasets are still valuable as they assess the performance of these tools using a common ground truth. To complement our existing analyses, we have assembled (using the same set of assembly methods) mouse sample replicates from LRGASP for three sequencing platforms: ONT cDNA, ONT dRNA, and PacBio CCS. Using SQANTI3 results for each assembly, we have generated Supplementary Figures 11, 12, and 13.

“Some aspects of the algorithm are unclear to me. How does digital normalization work? Will this normalization or other steps in the pipeline introduce any correction of the result? Does the method introduce any sequence correction based on long reads data?”

Response:

We have described in the Methods section (subsection “Digital normalization with strobemers”) detailing the algorithm we developed for digital normalization. As illustrated in Figure 1, digital normalization (stage 2) directly follows error correction (stage 1), which provides crude correction of sequencing errors. The polishing stage provides additional, high-resolution, sequence correction.

“One aspect of the long reads data we have observed is that tools report a high number of transcripts with one of few supporting reads. The RNA-bloom2 method evaluates the multiplicity of k-mers to identify strong and weak ones. Could you indicate which is the distribution of these multiplicity values and where the threshold is decided in each case? Supplementary figure 3 shows the concepts but not the actual values.”

Response:

The distribution of k -mer multiplicities is assumed to vary between tiles (from the same read or different reads) and that is the intention for the dynamic procedure illustrated in Supplementary Figure 4 (formerly Supplementary Figure 3). We have rephrased the caption for that figure to help clarify the procedure:

In each tile of a read, the k -mer multiplicity threshold for distinguishing strong and weak k -mers is dynamically set to the maximum of the fixed global threshold and the local threshold. The global threshold is defaulted to 2, unless specified otherwise by the ``-c`` option in RNA-Bloom2. To calculate the local threshold (M_{local}) for a tile, all k -mer multiplicity values within the tile are sorted in descending order and two consecutive values are evaluated at a time to search for the local threshold. When the second value is less than half of the first value, the local threshold is set to the first value and the search terminates. If no such local threshold is found, then the k -mer multiplicity threshold is set to the global threshold.

“When evaluating simulated and SIRVs data, How many total transcripts were detected? what is the distribution of transcripts per gene? Does the distribution match the true transcripts? Also, results should be broken down by accurate transcripts from multi-transcript genes and accurate transcripts from single transcript genes.”

Response:

We have generated Supplementary Tables 11 and 12, Supplementary Figures 9 and 10 to address these questions.

“What is the percentage of non-canonical splice sites present in the RNA-Bloom2 results?”

Response:

We have generated Supplementary Figure 14 for this metric based on the results from Supplementary Figures 11c, 12c, and 13c.

REVIEWERS' COMMENTS

Reviewer #1 (Remarks to the Author):

I appreciate the responses to all reviewers' comments, and the manuscript is substantially improved as a result of the revisions. I especially appreciate the additional supplementary materials including the analysis code newly made available. RNA-Bloom2 represents a clear advance for de novo reconstruction of transcripts from long reads and I wholeheartedly welcome it.

As per request, I examined responses to Reviewer #2's comments and conclude that the authors have addressed each satisfactorily, further improving the manuscript accordingly.

In re-reviewing the manuscript, I did have a couple of additional thoughts and comments that the authors may choose to address before publication.

- the importance of length normalization on unitigs for long reads (as opposed to short reads) is unclear. Read length normalization is done for short reads because longer transcripts produce more reads given the fragmentation of molecules that occurs prior to sequencing. Transcripts prepared for long reads aren't fragmented (fragmentation is avoided for long read sequencing, of course) and so it isn't obvious to me that similar transcript length normalization is relevant for long read transcriptome methods.

- after again studying main figure 1 part 6 I noticed that the transcript assembly illustration doesn't best represent the goal of extracting full length alternatively spliced isoforms, but instead looks more like teasing apart transcript contigs for different genes that might otherwise have irrelevant overlaps. If there's a way to better reflect isoform assembly in that figure, or better describe the main challenge being depicted in that part of the figure, that would be helpful.

Reviewer #3 (Remarks to the Author):

I would like to express my gratitude to the authors for addressing my concerns and incorporating additional evaluation metrics and procedures in the manuscript. I believe these additions provide a much clearer picture of RNA-Bloom2's performance.

However, I am still a bit uncertain about the scope and message of this paper. The work presents RNA-Bloom2 and provides extensive benchmarking and comparison to other approaches, such as RATTLE, which has similar goals, and reference-guided methods like StringTie2 and FLAIR. One would have expected this evaluation to result in clear statements on RNA-Bloom2's superior performance for reference-free assembly of transcripts using long reads, or at least an indication of when RNA-Bloom2 is a suitable choice. However, as presented, the paper reports benchmarking results in a descriptive manner and does not discuss them in detail in the Discussion, making it resemble more of a benchmarking paper. The Discussion section is limited to mentioning the comparison with RATTLE in terms of computational performance and discussing features of FLAIR and StringTie2, but it does not discuss performance insights obtained with the new metrics incorporated in this revision.

For instance, the results of the redundancy analysis (which are higher than RATTLE) and the higher number of FMS and ISM transcripts suggest that the resulting transcriptome might be "inflated," with multiple contigs representing the same true transcripts. Additionally, the high number of non-canonical splice junctions and NNC transcripts suggests that assembly inaccuracies may still populate the transcript models. The BUSCO analysis does not have the sensitivity to reveal this, as this analysis is biased toward recall. These features of the RNA-Bloom2 method should be discussed in terms of the possible limitations of the approach. Instead, the Discussion section only mentions RNA-Bloom2's ability to detect poly(A) signals and get the strand orientation in Nanopore reads right, without a real

assessment of this property. Furthermore, Supplementary figures 10-12 show a high level of antisense transcripts, so I am not convinced of this statement.

In summary, while I support the authors' discussion of the aspects where RNA-Bloom2 outperforms RATTLE and the contribution of the method to the field, I think the Discussion section should also include a fair discussion of the limitations, such as redundancy, partial transcripts, and inaccurate models.

Reference-free assembly of long-read transcriptome sequencing data with RNA-Bloom2

RESPONSE TO REVIEWERS' COMMENTS

Reviewer #1's remarks:

“- the importance of length normalization on unitigs for long reads (as opposed to short reads) is unclear. Read length normalization is done for short reads because longer transcripts produce more reads given the fragmentation of molecules that occurs prior to sequencing. Transcripts prepared for long reads aren't fragmented (fragmentation is avoided for long read sequencing, of course) and so it isn't obvious to me that similar transcript length normalization is relevant for long read transcriptome methods.”

Response:

We thank our reviewer for the comment. We have clarified this in the “Transcript assembly” section:

The length-normalized read counts for unitigs are measured as the number of aligned bases divided by the total length of the unitig. The length normalization is primarily intended to account for reads that align partially to more than one unitig, where a higher count is attributed to the unitig with more aligned bases. Without length normalization, read counts tallied from reads aligned to unitig overlaps would be double-counted, which is particularly detrimental when reads align to false-overlaps between unitigs. Therefore, the normalized read counts provide a means to discern false overlaps between unitigs that belong to transcripts with different expression level magnitudes.

“- after again studying main figure 1 part 6 I noticed that the transcript assembly illustration doesn't best represent the goal of extracting full length alternatively spliced isoforms, but instead looks more like teasing apart transcript contigs for different genes that might otherwise have irrelevant overlaps. If there's a way to better reflect isoform assembly in that figure, or better describe the main challenge being depicted in that part of the figure, that would be helpful.”

Response:

We thank our reviewer for the suggestion. We have revised Figure 1 to better reflect the assembly of alternative isoforms. In addition, we have expanded the figure legend to include a summary of the entire assembly workflow.

Reviewer #3's remarks:

“However, I am still a bit uncertain about the scope and message of this paper. The work presents RNA-Bloom2 and provides extensive benchmarking and comparison to other approaches, such as RATTLE, which has similar goals, and reference-guided methods like StringTie2 and FLAIR. One would have expected this evaluation to result in clear statements on RNA-Bloom2's superior performance for reference-free assembly of transcripts using long reads, or at least an indication of when RNA-Bloom2 is a suitable choice. However, as presented, the paper reports benchmarking results in a descriptive manner and does not discuss them in detail in the Discussion, making it resemble more of a benchmarking paper. The Discussion section is limited to mentioning the comparison with RATTLE in terms of computational performance and discussing features of FLAIR and StringTie2, but it does not discuss performance insights obtained with the new metrics incorporated in this revision.

For instance, the results of the redundancy analysis (which are higher than RATTLE) and the higher number of FMS and ISM transcripts suggest that the resulting transcriptome might be "inflated," with multiple contigs representing the same true transcripts. Additionally, the high number of non-canonical splice junctions and NNC transcripts suggests that assembly inaccuracies may still populate the transcript models. The BUSCO analysis does not have the sensitivity to reveal this, as this analysis is biased toward recall. These features of the RNA-Bloom2 method should be discussed in terms of the possible limitations of the approach. Instead, the Discussion section only mentions RNA-Bloom2's ability to detect poly(A) signals and get the strand orientation in Nanopore reads right, without a real assessment of this property. Furthermore, Supplementary figures 10-12 show a high level of antisense transcripts, so I am not convinced of this statement.

In summary, while I support the authors' discussion of the aspects where RNA-Bloom2 outperforms RATTLE and the contribution of the method to the field, I think the Discussion section should also include a fair discussion of the limitations, such as redundancy, partial transcripts, and inaccurate models.”

Response:

We thank the reviewer for the insight and recognizing the lack of discussion around these topics. We have extended two paragraphs in the Discussion section to discuss the aforementioned limitations.